# Joint Power Allocation for Coordinated Multi-Point Diversity Transmission in Rayleigh Fading Channels

**Min Lee [1] and Seong-Keun Oh [2],***

[1] Communication Networks, LIG Nex1 Co., Ltd., Seongnam 13488, Korea; minlee910@lignex1.com
[2] Department of Electrical and Computer Engineering, Ajou University, Suwon 16499, Korea
* Correspondence: oskn@ajou.ac.kr; Tel.: +82-31-219-2370

**Abstract:** We consider the problem of joint power allocation (JPA) in a coordinated multi-point (CoMP) joint diversity transmission (JDT) network with a total coordination point power (TCPP) constraint, aimed at maximizing the ergodic cooperative capacity (ECC) in Rayleigh fading channels. In this paper, we first extend the JPA problem in the coordinated two-point (Co2P) JDT network to the case of a non-unity TCPP constraint. Furthermore, we introduce more accurate log-quadratic approximated (LQA) expressions to obtain the coordinated transmission point (CTP) powers. Next, we extend our study to a coordinated three-point (Co3P) JDT network. Given the mean branch gain-to-noise ratios, we first obtain a log-linear approximated (LLA) expression for obtaining the optimum power of the third CTP (i.e., the worst quality-providing CTP). After obtaining the third-CTP power, we obtain the CTP powers of two better quality-providing CTPs by invoking the LLA CTP power expressions for Co2P JDT power allocation, under the remaining power given by the TCPP minus the third-CTP power. The numerical results demonstrate that the LQA and LLA CTP power expressions for Co2P JDT and the LLA CTP power expressions for Co3P JDT are very efficient in terms of the simplicity for JPA and CoMP set selection, as well as the resulting ECC.

**Keywords:** CoMP network; joint diversity transmission; joint power allocation; total coordination point power constraint; ergodic cooperative capacity

## 1. Introduction

In a cellular network, the cell-edge user throughput is mainly limited owing to the path loss by the long distance from the serving base station and the inter-cell interference from neighboring base stations. Coordinated multi-point (CoMP) transmission technology can improve the signal quality and/or throughput at a cell-edge region and can also widen the coverage of high throughput via cooperative transmission and/or coordinated inter-cell interference control by multiple coordinated transmission points (CTPs) [1,2]. CoMP has been included not only as a promising technology for fourth-generation (4G) mobile communication standards, but also it is expected to suit even better in fifth-generation networks, whose novel architectures will accommodate extremely larger amount of data traffic that can be currently circulated in 4G networks [1]. In particular, CoMP joint diversity transmission (JDT) can significantly improve the signal quality because two or more CTPs coordinately transmit signals so as to be meaningfully exploited at a user equipment (UE) receiver, at the expense of a radio-resource deficiency [2].

Many studies on CoMP transmission technology have been conducted [3–6]. For CoMP transmission technology, expanded researches have been conducted recently such as the joint transmission techniques with adaptive modulation [3], spectrum resource allocation method for CoMP transmissions [4], as well as fundamental researches have been conducted steadily such as the coordinated beamforming techniques [5], the determination techniques of a CTP set [6]. In the context

of wireless green communication [7], the total coordination point power (TCPP) over all CTPs in the cooperating set for a targeted UE could be constrained to a certain level [8–10]. Hence, the CTPs should share the TCPP by distributing it efficiently. However, in a CoMP configuration where all the CTPs in the cooperating set for the targeted UE share the TCPP, JDT may not be effective if the received signal level from one CTP is extremely lower or higher than those from the other CTPs. Therefore, a simple and efficient process to determine the number of CTPs participating in JDT and subsequently, the power levels to be allocated over the CTPs in the resulting JDT cooperating set should be developed.

In [11], an optimum joint power allocation (JPA) scheme to maximize the instantaneous capacity in an orthogonal frequency division multiple access CoMP JDT network with individual CTP power constraints was introduced. However, this scheme may be used only in a slowly varying fading environment because the instantaneous channel state information may not be exchanged quickly enough among the CTPs. In [8], the authors dealt with the problem of JPA in a coordinated two-point (Co2P) JDT network with a unity TCPP constraint, aimed at maximizing the ergodic cooperative capacity (ECC) in Rayleigh fading channels. However, the approach in [8] cannot be used in a CoMP configuration when two CTPs in the cooperating set share a non-unity TCPP and/or in a CoMP configuration where more than two CTPs participate in JDT under a TCPP constraint.

In this paper, we consider the problem of JPA in a CoMP JDT network with a TCPP constraint, to maximize the ECC in Rayleigh fading channels. We first extend the JDT power allocation problem in the Co2P JDT network with the unity TCPP constraint in [8] to the case of a non-unity TCPP constraint. Furthermore, we introduce more accurate log-quadratic approximated (LQA) and log-linear approximated (LLA) CTP power expressions with respect to the mean branch gain-to-noise ratios (GNRs). Next, we extend our study to a coordinated three-point (Co3P) JDT network with the relationship that yields the optimum three CTP powers as a function of the mean branch GNRs. Since the relationship contains several complicated operations such as exponentials, exponential integrals and their products, we derive an LLA expression to obtain the optimum power for the third CTP (i.e., the worst quality-providing CTP, hereafter referred to as CTP3 in the Co3P JDT network) with respect to the mean CTP3-UE branch GNR. This LLA expression connects the CTP3 powers corresponding to two different mean CTP3-UE branch GNR values: the mean CTP3-UE branch GNR value dividing the Co3P JDT mode and the Co2P JDT mode by the two better quality-providing CTPs, and the settled point where the mean CTP3-UE branch GNR is equal to the mean median-CTP (i.e., the median quality-providing CTP, hereafter referred to as CTP2 in the Co3P JDT network) branch GNR (i.e., the mean CTP2-UE branch GNR). After obtaining the approximate third-CTP power, we obtain the CTP powers of the two better quality-providing CTPs (i.e., CTP1 and CTP2, with CTP1 denoting the best quality-providing CTP) by using the LLA CTP power expressions used for Co2P JDT power allocation, under the remaining power given by the TCPP minus the third-CTP power. Finally, we perform numerical analyses to demonstrate the ECC efficiencies when the LQA and LLA CTP power expressions for Co2P JDT and the LLA CTP power expressions for Co3P JDT are used.

## 2. CoMP JDT Network

We consider a CoMP JDT network where two or more CTPs in a JDT cooperating set, each CTP equipped with a single antenna coordinately transmit signals to a targeted UE equipped with a single antenna, as shown in Figure 1a,b. We assume that the CTPs in the cooperating set share data and mean channel quality indicators for the target UE via reliable backhaul connections. In addition, the CTPs in the cooperating set share a TCPP $P_S$ such as $P_S = P_1 + P_2$ for Co2P JDT and $P_S = P_1 + P_2 + P_3$ for Co3P JDT, where $P_1$, $P_2$, and $P_3$ denote the respective CTP powers allocated to CTP1, CTP2, and CTP3.

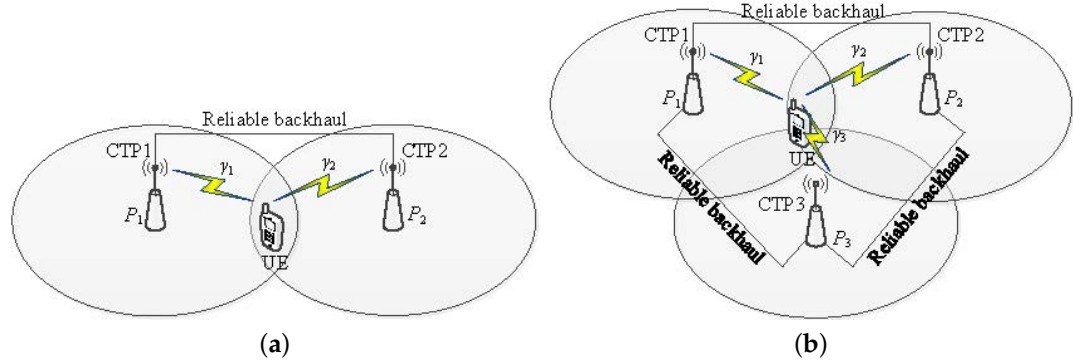

**Figure 1.** CoMP JDT networks: (**a**) Co2P case (**b**) Co3P case.

The respective instantaneous diversity-combined signal-to-noise ratios at the UE receiver, for Co2P JDT and Co3P JDT are given by [12]

$$\gamma_{co2} = P_1\gamma_1 + P_2\gamma_2, \tag{1}$$

$$\gamma_{co3} = P_1\gamma_1 + P_2\gamma_2 + P_3\gamma_3, \tag{2}$$

where $\gamma_1$, $\gamma_2$, and $\gamma_3$ denote the CTP1-UE, CTP2-UE, and CTP3-UE branch GNRs, respectively. In Rayleigh fading channels, $\gamma_1$, $\gamma_2$, and $\gamma_3$ are independent exponential random variables with the respective means of $\bar{\gamma}_1$, $\bar{\gamma}_2$, and $\bar{\gamma}_3$.

## 3. Joint Power Allocation for Co2P JDT under a Non-Unity TCPP

We formulate the problem of JPA for Co2P JDT and Co3P JDT under a TCPP constraint, to maximize the ECC in Rayleigh fading channels. This problem can be expressed as

$$\max_{P_i} \quad \bar{C}_{jdt}, \tag{3a}$$

$$\text{s.t.} \quad \sum_{i=1}^{M} P_i \leq P_S, \tag{3b}$$

$$0 \leq P_i \leq P_S, \ i = 1, \cdots, M, \tag{3c}$$

where $\bar{C}_{jdt}$ denotes the ECC achieved by the CoMP JDT network; and $M$ denotes the number of CTPs in the JDT cooperating set. It should be noted that $\bar{C}_{jdt}$ is the single-branch ergodic capacity in the case of non-cooperative transmission (i.e., non-JDT mode) and the ECC in the case of cooperative transmission (i.e., CoMP JDT mode). In Rayleigh fading channels, the instantaneous diversity-combined signal-to-noise ratio in Equation (1) is the sum of two independent exponential random variables $P_1\gamma_1$ and $P_2\gamma_2$. Hence, the probability density function by the Co2P JDT network can be expressed as

$$f_{\Gamma_{co2}}(\gamma_{co2}) = \begin{cases} \dfrac{e^{-\frac{\gamma_{co2}}{P_1\bar{\gamma}_1}} - e^{-\frac{\gamma_{co2}}{P_2\bar{\gamma}_2}}}{P_1\bar{\gamma}_1 - P_2\bar{\gamma}_2}, & \text{if } P_1\bar{\gamma}_1 \neq P_2\bar{\gamma}_2, \\ \dfrac{\gamma_{co2}}{P_1^2\bar{\gamma}_1^2} e^{-\frac{\gamma_{co2}}{P_1\bar{\gamma}_1}}, & \text{if } P_1\bar{\gamma}_1 = P_2\bar{\gamma}_2, \end{cases} \tag{4}$$

Then, the ECC achieved by the Co2P JDT network is given by Equation (5) [8]

$$C_{2,jdt} = \begin{cases} \dfrac{\log_2(e)}{P_1\bar{\gamma}_1 - P_2\bar{\gamma}_2} \left\{ P_1\bar{\gamma}_1 e^{\frac{1}{P_1\bar{\gamma}_1}} E_1\left(\frac{1}{P_1\bar{\gamma}_1}\right) - P_2\bar{\gamma}_2 e^{\frac{1}{P_2\bar{\gamma}_2}} E_1\left(\frac{1}{P_2\bar{\gamma}_2}\right) \right\}, & \text{if } P_1\bar{\gamma}_1 \neq P_2\bar{\gamma}_2, \\ \log_2(e) \left\{ 1 - \left(\frac{1}{P_1\bar{\gamma}_1} - 1\right) e^{\frac{1}{P_1\bar{\gamma}_1}} E_1\left(\frac{1}{P_1\bar{\gamma}_1}\right) \right\}, & \text{if } P_1\bar{\gamma}_1 = P_2\bar{\gamma}_2, \end{cases} \tag{5}$$

where $E_1(\cdot)$ denotes the first-order exponential integral function [13].

To calculate the optimum CTP powers to maximize the ECC, substituting $P_1 = P_S - P_2$ into the top of Equation (5), differentiating the resulting ECC expression with respect to $P_2$, and setting the derivative to zero, we obtain the following expression:

$$
\begin{aligned}
&\frac{\log_2(e)}{(P_S - P_2^*)\,\bar{\gamma}_1 - P_2^*\bar{\gamma}_2} \left\{ \left( \bar{\gamma}_2 - \frac{1}{P_2^*} \right) e^{\frac{1}{P_2^*\bar{\gamma}_2}} E_1\left( \frac{1}{P_2^*\bar{\gamma}_2} \right) \right. \\
&\left. + \left( \bar{\gamma}_1 - \frac{1}{P_S - P_2^*} \right) e^{\frac{1}{(P_S - P_2^*)\bar{\gamma}_1}} E_1\left( \frac{1}{(P_S - P_2^*)\,\bar{\gamma}_1} \right) + \bar{\gamma}_1 + \bar{\gamma}_2 \right\} \\
&+ \frac{(\bar{\gamma}_1 + \bar{\gamma}_2)\log_2(e)}{\left\{ (P_S - P_2^*)\,\bar{\gamma}_1 - P_2^*\bar{\gamma}_2 \right\}^2} \left\{ P_2^*\bar{\gamma}_2 e^{\frac{1}{P_2^*\bar{\gamma}_2}} E_1\left( \frac{1}{P_2^*\bar{\gamma}_2} \right) \right. \\
&\left. - (P_S - P_2^*)\,\bar{\gamma}_1 e^{\frac{1}{(P_S - P_2^*)\bar{\gamma}_1}} E_1\left( \frac{1}{(P_S - P_2^*)\,\bar{\gamma}_1} \right) \right\} = 0.
\end{aligned}
\tag{6}
$$

Here, $P_1^*$ and $P_2^*$ denote the respective optimum CTP powers to be allocated to CTP1 and CTP2 for Co2P JDT. Because Equation (6) contains several complicated operations such as exponentials, exponential integrals, and their products, we can solve Equation (6) by only using numerical techniques. Thus, the optimum CTP powers for Co2P JDT might not be used in practice owing to the high complexity of implementation.

At this point, we introduce a much simpler CTP power expression for use in practice. The solid lines in Figure 2a–c show $P_2^*$ versus $\bar{\gamma}_2$[dB], with $\bar{\gamma}_1$[dB] as a parameter, when $P_S$ = 2, 4, and 8 W, respectively. In this study, we obtained the solid lines in Figure 2a–c from Equation (6) by using the bisection method [14]. From these solid lines, we see that in the condition $\bar{\gamma}_1 \geq \bar{\gamma}_2$, $P_2^*$ is almost quadratic with respect to $\bar{\gamma}_2$[dB] over the corresponding Co2P JDT region (i.e., the region from the boundary point dividing the Co2P JDT mode and the non-JDT mode, $\bar{\gamma}_2^{bd}$ to the balanced point of $P_2^* = P_1^*$ at $\bar{\gamma}_2 = \bar{\gamma}_1$). Therefore, we use an LQA CTP power expression for $P_2^*$ versus $\bar{\gamma}_2$[dB], which is a quadratic line connecting the respective optimum CTP2 powers at the boundary and balanced points, while satisfying the slope at the balanced point.

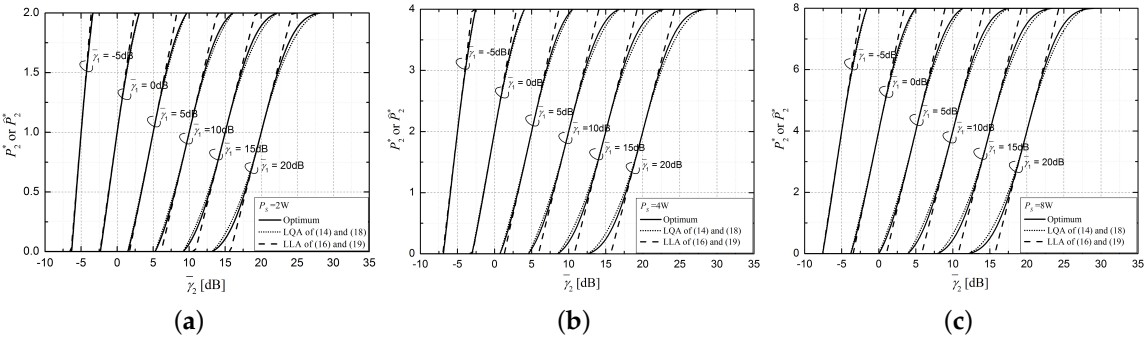

**Figure 2.** $P_2^*$ versus $\bar{\gamma}_2$[dB], with $\bar{\gamma}_1$[dB] as a parameter: (**a**) $P_S$ = 2 W, (**b**) $P_S$ = 4 W, (**c**) $P_S$ = 8 W.

To derive the LQA CTP power expression in the condition $\bar{\gamma}_1 \geq \bar{\gamma}_2$, we first derive the boundary point. Given $\bar{\gamma}_1$, we obtain Equation (7) after substituting the boundary point condition (i.e., $P_2^* = 0$) into Equation (6).

$$
\begin{aligned}
&-\frac{1}{P_S \bar{\gamma}_1} \left\{ \lim_{P_2^* \to 0} \bar{\gamma}_2^{bd} e^{\frac{1}{P_2^* \bar{\gamma}_2^{bd}}} E_1\left(\frac{1}{P_2^* \bar{\gamma}_2^{bd}}\right) - \lim_{P_2^* \to 0} \frac{1}{P_2^*} e^{\frac{1}{P_2^* \bar{\gamma}_2^{bd}}} E_1\left(\frac{1}{P_2^* \bar{\gamma}_2^{bd}}\right) \right. \\
&\left. + \left(\bar{\gamma}_1 - \frac{1}{P_S}\right) e^{\frac{1}{P_S \bar{\gamma}_1}} E_1\left(\frac{1}{P_S \bar{\gamma}_1}\right) + \bar{\gamma}_2^{bd} + \bar{\gamma}_1 \right\} \\
&+ \frac{\bar{\gamma}_2^{bd} + \bar{\gamma}_1}{(P_S \bar{\gamma}_1)^2} \left\{ P_S \bar{\gamma}_1 e^{\frac{1}{P_S \bar{\gamma}_1}} E_1\left(\frac{1}{P_S \bar{\gamma}_1}\right) - \lim_{P_2^* \to 0} P_2^* \bar{\gamma}_2^{bd} e^{\frac{1}{P_2^* \bar{\gamma}_2^{bd}}} E_1\left(\frac{1}{P_2^* \bar{\gamma}_2^{bd}}\right) \right\} = 0.
\end{aligned}
\tag{7}
$$

Invoking the L'Hopital's rule, we obtain the following three limiting terms as

$$
\lim_{P_2^* \to 0} \bar{\gamma}_2^{bd} e^{\frac{1}{P_2^* \bar{\gamma}_2^{bd}}} E_1\left(\frac{1}{P_2^* \bar{\gamma}_2^{bd}}\right) = 0,
\tag{8a}
$$

$$
\lim_{P_2^* \to 0} \frac{1}{P_2^*} e^{\frac{1}{P_2^* \bar{\gamma}_2^{bd}}} E_1\left(\frac{1}{P_2^* \bar{\gamma}_2^{bd}}\right) = \bar{\gamma}_2^{bd},
\tag{8b}
$$

$$
\lim_{P_2^* \to 0} P_2^* \bar{\gamma}_2^{bd} e^{\frac{1}{P_2^* \bar{\gamma}_2^{bd}}} E_1\left(\frac{1}{P_2^* \bar{\gamma}_2^{bd}}\right) = 0.
\tag{8c}
$$

Substituting the three limiting terms in Equation (8) into Equation (7) and rearranging the terms, we obtain the boundary point as

$$
\bar{\gamma}_2^{bd} = \frac{\bar{\gamma}_1}{e^{\frac{1}{P_S \bar{\gamma}_1}} E_1\left(\frac{1}{P_S \bar{\gamma}_1}\right)} - \frac{1}{P_S}.
\tag{9}
$$

Next, we derive the slope of $P_2^*$ with respect to $\bar{\gamma}_2$[dB] at the balanced point. For ease of derivation, we set $\alpha_2 = \ln(\bar{\gamma}_2)$. By differentiating Equation (6) with respect to $\alpha_2$, after some manipulations, we obtain Equation (10).

$$
\frac{dP_2^*}{d\alpha_2} = \frac{N_{T'}}{D_{T'}},
\tag{10}
$$

with

$$
\begin{aligned}
N_{T'} = & \left\{ P_S e^{\alpha_2} \bar{\gamma}_1 + e^{\alpha_2} - \frac{1}{P_2^*} - \frac{P_S \bar{\gamma}_1}{P_2^*} + \frac{P_S \bar{\gamma}_1}{P_2^{*2} e^{\alpha_2}} - \frac{\bar{\gamma}_1}{P_2^* e^{\alpha_2}} \right\} e^{\frac{1}{P_2^* e^{\alpha_2}}} E_1\left(\frac{1}{P_2^* e^{\alpha_2}}\right) \\
& + \left(\frac{P_2^* e^{\alpha_2}}{P_S - P_2^*} - P_S e^{\alpha_2} \bar{\gamma}_1\right) e^{\frac{1}{(P_S - P_2^*) \bar{\gamma}_1}} E_1\left(\frac{1}{(P_S - P_2^*) \bar{\gamma}_1}\right) \\
& - \{P_2^* e^{\alpha_2} - (P_S - P_2^*) \bar{\gamma}_1\} \left(2 e^{\alpha_2} - \frac{1}{P_2^*}\right),
\end{aligned}
$$

$$
\begin{aligned}
D_{T'} = & \{P_2^* e^{\alpha_2} - (P_S - P_2^*) \bar{\gamma}_1\} \left\{ \frac{1}{P_2^{*3} e^{\alpha_2}} e^{\frac{1}{P_2^* e^{\alpha_2}}} E_1\left(\frac{1}{P_2^* e^{\alpha_2}}\right) \right. \\
& - \frac{1}{(P_S - P_2^*)^3 \bar{\gamma}_1} e^{\frac{1}{(P_S - P_2^*) \bar{\gamma}_1}} E_1\left(\frac{1}{(P_S - P_2^*) \bar{\gamma}_1}\right) \\
& \left. + \frac{1}{P_2^*} \left(e^{\alpha_2} - \frac{1}{P_2^*}\right) - \frac{1}{P_S - P_2^*} \left(\bar{\gamma}_1 - \frac{1}{P_S - P_2^*}\right) \right\}.
\end{aligned}
$$

However, we cannot evaluate the slope $dP_2^*/d\alpha_2|_{\bar{\gamma}_2=\bar{\gamma}_1,P_2^*=0.5P_S}$ because both the numerator and denominator on the right-hand side of Equation (10) are zero at the balanced point. According to the L'Hopital's rule, by taking the respective second-order derivatives of both the numerator and denominator with respect to $\alpha_2$ and substituting the balanced point condition such as $P_2^* = P_1^* = 0.5P_S$ at $\bar{\gamma}_2 = \bar{\gamma}_1$ into their respective second derivatives, we obtain Equation (11).

$$\lim_{\bar{\gamma}_2\to\bar{\gamma}_1,P_2^*\to0.5P_S}\frac{dP_2^*}{d\alpha_2} = \lim_{\bar{\gamma}_2\to\bar{\gamma}_1,P_2^*\to0.5P_S}\frac{d^2N_{T'}/d\alpha_2{}^2}{d^2D_{T'}/d\alpha_2{}^2} = \frac{N_{\beta'}}{D_{\beta'}}, \tag{11}$$

with

$$
\begin{aligned}
N_{\beta'} = & \left\{ \left(\frac{dP_2^*}{d\alpha_2}\right)^2\frac{64}{P_S^3} + \left(\frac{dP_2^*}{d\alpha_2}\right)^2\frac{64}{P_S^4\bar{\gamma}_1} + \frac{dP_2^*}{d\alpha_2}\frac{16}{P_S^2} + \frac{dP_2^*}{d\alpha_2}\frac{32}{P_S^3\bar{\gamma}_1} + \frac{4}{P_S^2\bar{\gamma}_1} \right\} \\
& \cdot e^{\frac{2}{P_S\bar{\gamma}_1}}E_1\left(\frac{2}{P_S\bar{\gamma}_1}\right) - \left(\frac{dP_2^*}{d\alpha_2}\right)^2\frac{16\bar{\gamma}_1}{P_S^2} - \left(\frac{dP_2^*}{d\alpha_2}\right)^2\frac{32}{P_S^3} \\
& - \frac{dP_2^*}{d\alpha_2}\frac{16}{P_S^2} - \frac{dP_2^*}{d\alpha_2}4\bar{\gamma}_1^2 - P_S\bar{\gamma}_1^2 + \bar{\gamma}_1 - \frac{2}{P_S},
\end{aligned}
$$

$$
\begin{aligned}
D_{\beta'} = & -\left\{ \left(\frac{dP_2^*}{d\alpha_2}\right)^2\frac{384}{P_S^4} + \left(\frac{dP_2^*}{d\alpha_2}\right)^2\frac{256}{P_S^5\bar{\gamma}_1} + \frac{dP_2^*}{d\alpha_2}\frac{128}{P_S^3} + \frac{dP_2^*}{d\alpha_2}\frac{128}{P_S^4\bar{\gamma}_1} + \frac{8}{P_S^2} + \frac{16}{P_S^3\bar{\gamma}_1} \right\} \\
& \cdot e^{\frac{2}{P_S\bar{\gamma}_1}}E_1\left(\frac{2}{P_S\bar{\gamma}_1}\right) + \left(\frac{dP_2^*}{d\alpha_2}\right)^2\frac{128}{P_S^4} + \left(\frac{dP_2^*}{d\alpha_2}\right)^2\frac{128\bar{\gamma}_1}{P_S^3} - \left(\frac{dP_2^*}{d\alpha_2}\right)^2\frac{32\bar{\gamma}_1^2}{P_S^2} \\
& + \frac{dP_2^*}{d\alpha_2}\frac{64}{P_S^3} + \frac{dP_2^*}{d\alpha_2}\frac{32\bar{\gamma}_1}{P_S^2} + \frac{8}{P_S^2} + 2\bar{\gamma}_1^2.
\end{aligned}
$$

From Equation (11) and by defining $\beta' = dP_2^*/d\alpha_2|_{\bar{\gamma}_2=\bar{\gamma}_1,P_2^*=0.5P_S}$, we obtain Equation (12) of the cubic polynomial of $\beta'$ after factorization.

$$
\begin{aligned}
& -\frac{1}{\bar{\gamma}_1}\left(\frac{4\beta'}{P_S}+1\right)^2\left[\beta'\left\{\left(\frac{24\bar{\gamma}_1}{P_S^2}+\frac{16}{P_S^3}\right)e^{\frac{2}{P_S\bar{\gamma}_1}}E_1\left(\frac{2}{P_S\bar{\gamma}_1}\right)+2\bar{\gamma}_1^3-\frac{8\bar{\gamma}_1^2}{P_S}-\frac{8\bar{\gamma}_1}{P_S^2}\right\}\right. \\
& \left. +\frac{4}{P_S^2}e^{\frac{2}{P_S\bar{\gamma}_1}}E_1\left(\frac{2}{P_S\bar{\gamma}_1}\right)-P_S\bar{\gamma}_1^3-\frac{2\bar{\gamma}_1}{P_S}+\bar{\gamma}_1^2\right] = 0. \tag{12}
\end{aligned}
$$

From the solid lines in Figure 2a–c, we note that $\beta'$ is a positive real value. Therefore, the valid slope of $P_2^*$ with respect to $\alpha_2$ at the balanced point, $\beta' = \beta'(\bar{\gamma}_1)$ is

$$\beta'(\bar{\gamma}_1) = \frac{-\frac{4}{P_S^2}e^{\frac{2}{P_S\bar{\gamma}_1}}E_1\left(\frac{2}{P_S\bar{\gamma}_1}\right)+P_S\bar{\gamma}_1^3+\frac{2\bar{\gamma}_1}{P_S}-\bar{\gamma}_1^2}{\left(\frac{24\bar{\gamma}_1}{P_S^2}+\frac{16}{P_S^3}\right)e^{\frac{2}{P_S\bar{\gamma}_1}}E_1\left(\frac{2}{P_S\bar{\gamma}_1}\right)+2\bar{\gamma}_1^3-\frac{8\bar{\gamma}_1^2}{P_S}-\frac{8\bar{\gamma}_1}{P_S^2}}. \tag{13}$$

The slope of $P_2^*$ with respect to $\bar{\gamma}_2[\mathrm{dB}]$ at the balanced point, $\hat{\beta}(\bar{\gamma}_1)$ is given by $\hat{\beta}(\bar{\gamma}_1) = \beta'(\bar{\gamma}_1)\ln 10/10 = dP_2^*/d\bar{\gamma}_2[\mathrm{dB}]|_{\bar{\gamma}_2=\bar{\gamma}_1,P_2^*=0.5P_S}$.

To obtain the LQA CTP2 power expression, we express the quadratic equation for $P_2^*$ versus $\bar{\gamma}_2[\mathrm{dB}]$ as $P_2^{q*} = A\bar{\gamma}_2[\mathrm{dB}]^2 + B\bar{\gamma}_2[\mathrm{dB}] + C$. This quadratic equation must meet the following three conditions: $P_2^{q*} = 0$ at $\bar{\gamma}_2 = \bar{\gamma}_2^{bd}$ (i.e., the boundary point condition), $P_2^{q*} = 0.5P_S$ at $\bar{\gamma}_2 = \bar{\gamma}_1$ (i.e., the balanced point condition), and the slope at the balanced point $\hat{\beta}(\bar{\gamma}_1)$. Using these three conditions, we obtain three simultaneous equations with respect to $A$, $B$, and $C$:

$$A\bar{\gamma}_2^{bd}[\mathrm{dB}]^2 + B\bar{\gamma}_2^{bd}[\mathrm{dB}] + C = 0, \tag{14a}$$

$$A\bar{\gamma}_1[\text{dB}]^2 + B\bar{\gamma}_1[\text{dB}] + C = 0.5P_S, \tag{14b}$$

$$2A\bar{\gamma}_1[\text{dB}] + B = \hat{\beta}(\bar{\gamma}_1). \tag{14c}$$

Solving these three simultaneous equations, we obtain Equation (15) of the LQA CTP power expression for $P_2^*$ versus $\bar{\gamma}_2[\text{dB}]$.

$$
\begin{aligned}
P_2^{q*} &= A\bar{\gamma}_2[\text{dB}]^2 + B\bar{\gamma}_2[\text{dB}] + C \\
&= \left\{ -\frac{\hat{\beta}(\bar{\gamma}_1)}{\bar{\gamma}_2^{bd}[\text{dB}] - \bar{\gamma}_1[\text{dB}]} - \frac{0.5P_S}{\left(\bar{\gamma}_2^{bd}[\text{dB}] - \bar{\gamma}_1[\text{dB}]\right)^2} \right\} \bar{\gamma}_2[\text{dB}]^2 \\
&+ \left\{ \hat{\beta}(\bar{\gamma}_1) + \frac{2\hat{\beta}(\bar{\gamma}_1)\bar{\gamma}_1[\text{dB}]}{\bar{\gamma}_2^{bd}[\text{dB}] - \bar{\gamma}_1[\text{dB}]} + \frac{P_S\bar{\gamma}_1[\text{dB}]}{\left(\bar{\gamma}_2^{bd}[\text{dB}] - \bar{\gamma}_1[\text{dB}]\right)^2} \right\} \bar{\gamma}_2[\text{dB}] \\
&+ \left\{ \frac{\hat{\beta}(\bar{\gamma}_1)}{\bar{\gamma}_2^{bd}[\text{dB}] - \bar{\gamma}_1[\text{dB}]} + \frac{0.5P_S}{\left(\bar{\gamma}_2^{bd}[\text{dB}] - \bar{\gamma}_1[\text{dB}]\right)^2} \right\} \bar{\gamma}_2^{bd}[\text{dB}]^2 \\
&- \left\{ \hat{\beta}(\bar{\gamma}_1) + \frac{2\hat{\beta}(\bar{\gamma}_1)\bar{\gamma}_1[\text{dB}]}{\bar{\gamma}_2^{bd}[\text{dB}] - \bar{\gamma}_1[\text{dB}]} + \frac{P_S\bar{\gamma}_1[\text{dB}]}{\left(\bar{\gamma}_2^{bd}[\text{dB}] - \bar{\gamma}_1[\text{dB}]\right)^2} \right\} \bar{\gamma}_2^{bd}[\text{dB}].
\end{aligned} \tag{15}
$$

In the condition $\bar{\gamma}_1 \geq \bar{\gamma}_2$, we calculate the CTP2 power to be actually allocated to CTP2, $\hat{P}_2^*$ by using the LQA CTP2 power expression of Equation (15) as

$$\hat{P}_2^* = \begin{cases} 0, & \text{if } P_2^{q*} < 0, \\ P_2^{q*}, & \text{if } P_2^{q*} \geq 0. \end{cases} \tag{16}$$

Then, we can obtain the CTP1 power to be actually allocated to CTP1, $\hat{P}_1^*$ as $\hat{P}_1^* = P_S - \hat{P}_2^*$.

Furthermore, using $\beta'(\bar{\gamma}_1)$, we obtain the LLA CTP2 power expression similar to Equation (12) of Ref. [8] as

$$P_2^{l*} = \beta'(\bar{\gamma}_1)\left(\ln\bar{\gamma}_2 - \ln\bar{\gamma}_1\right) + 0.5P_S. \tag{17}$$

In the condition $\bar{\gamma}_1 \geq \bar{\gamma}_2$, we calculate the CTP2 power to be actually allocated to CTP2, $\hat{P}_2^*$ by using the LLA CTP2 power expressions of (17) as

$$\hat{P}_2^* = \begin{cases} 0, & \text{if } P_2^{l*} < 0, \\ P_2^{l*}, & \text{if } P_2^{l*} \geq 0. \end{cases} \tag{18}$$

Then, we again can obtain the CTP1 power to be actually allocated to CTP1, $\hat{P}_1^*$ as $\hat{P}_1^* = P_S - \hat{P}_2^*$.

In conclusion, we can decide whether Co2P JDT is performed or not by using Equation (16) or Equation (18). In the condition $\bar{\gamma}_1 \geq \bar{\gamma}_2$, Co2P JDT is performed only if $\hat{P}_2^* > 0$; single-point transmission by CTP1 is performed if $\hat{P}_2^* = 0$.

In the condition $\bar{\gamma}_1 < \bar{\gamma}_2$, we can obtain Equation (19) of the LQA CTP power expression for $P_1^*$ versus $\bar{\gamma}_1[\text{dB}]$ instead of $P_2^*$ versus $\bar{\gamma}_2[\text{dB}]$ by exchanging the roles of CTP1 and CTP2, that is, $P_2^{q*} \to P_1^{q*}$, $\bar{\gamma}_2[\text{dB}] \to \bar{\gamma}_1[\text{dB}]$, $\bar{\gamma}_1[\text{dB}] \to \bar{\gamma}_2[\text{dB}]$, $\bar{\gamma}_2^{bd}[\text{dB}] \to \bar{\gamma}_1^{bd}[\text{dB}]$, and $\hat{\beta}(\bar{\gamma}_1) \to \hat{\beta}(\bar{\gamma}_2)$ in (15).

$$
\begin{aligned}
P_1^{q*} = \;& \left\{ -\frac{\hat{\beta}\left(\bar{\gamma}_2\right)}{\bar{\gamma}_1^{bd}[\text{dB}] - \bar{\gamma}_2[\text{dB}]} - \frac{0.5P_S}{\left(\bar{\gamma}_1^{bd}[\text{dB}] - \bar{\gamma}_2[\text{dB}]\right)^2} \right\} \bar{\gamma}_1[\text{dB}]^2 \\
& + \left\{ \hat{\beta}\left(\bar{\gamma}_2\right) + \frac{2\hat{\beta}\left(\bar{\gamma}_2\right)\bar{\gamma}_2[\text{dB}]}{\bar{\gamma}_1^{bd}[\text{dB}] - \bar{\gamma}_2[\text{dB}]} + \frac{P_S\bar{\gamma}_2[\text{dB}]}{\left(\bar{\gamma}_1^{bd}[\text{dB}] - \bar{\gamma}_2[\text{dB}]\right)^2} \right\} \bar{\gamma}_1[\text{dB}] \\
& + \left\{ \frac{\hat{\beta}\left(\bar{\gamma}_2\right)}{\bar{\gamma}_1^{bd}[\text{dB}] - \bar{\gamma}_2[\text{dB}]} + \frac{0.5P_S}{\left(\bar{\gamma}_1^{bd}[\text{dB}] - \bar{\gamma}_2[\text{dB}]\right)^2} \right\} \bar{\gamma}_1^{bd}[\text{dB}]^2 \\
& - \left\{ \hat{\beta}\left(\bar{\gamma}_2\right) + \frac{2\hat{\beta}\left(\bar{\gamma}_2\right)\bar{\gamma}_2[\text{dB}]}{\bar{\gamma}_1^{bd}[\text{dB}] - \bar{\gamma}_2[\text{dB}]} + \frac{P_S\bar{\gamma}_2[\text{dB}]}{\left(\bar{\gamma}_1^{bd}[\text{dB}] - \bar{\gamma}_2[\text{dB}]\right)^2} \right\} \bar{\gamma}_1^{bd}[\text{dB}].
\end{aligned}
\tag{19}
$$

We also obtain the LLA CTP power expression for $P_1^*$ versus $\bar{\gamma}_1[\text{dB}]$ as follows:

$$
P_1^{l*} = \beta'\left(\bar{\gamma}_2\right)\left(\ln \bar{\gamma}_1 - \ln \bar{\gamma}_2\right) + 0.5P_S.
\tag{20}
$$

In Figure 2a–c, the dotted and dashed lines, respectively, indicate the CTP2 powers when the LQA and LLA CTP power expressions are used. From Figure 2a–c, we see that the CTP2 power using the LQA CTP power expression is closer to the corresponding optimum CTP power over a wider Co2P JDT region around the balanced point than that using the LLA CTP power expression.

## 4. Joint Power Allocation for Co3P JDT

In this section, we extend our study to the Co3P JDT network with a non-unity TCPP constraint. In Rayleigh fading channels, the ECC achieved by the Co3P JDT network is given by Equation (21) [15].

$$
\bar{C}_{3,jdt} = \begin{cases}
\log_2\left(e\right) \sum\limits_{i=1}^{3} \left\{ \prod\limits_{j=1, j\neq i}^{3} \left( \frac{P_i\bar{\gamma}_i}{P_i\bar{\gamma}_i - P_j\bar{\gamma}_j} \right) e^{\frac{1}{P_i\bar{\gamma}_i}} E_1\left( \frac{1}{P_i\bar{\gamma}_i} \right) \right\}, & \text{if } P_1\bar{\gamma}_1 \neq P_2\bar{\gamma}_2 \neq P_3\bar{\gamma}_3, \\[3mm]
\frac{P_1\bar{\gamma}_1\log_2(e)}{(P_3\bar{\gamma}_3 - P_1\bar{\gamma}_1)^2}\left\{ P_1\bar{\gamma}_1 e^{\frac{1}{P_1\bar{\gamma}_1}} E_1\left( \frac{1}{P_1\bar{\gamma}_1} \right) - P_3\bar{\gamma}_3 e^{\frac{1}{P_3\bar{\gamma}_3}} E_1\left( \frac{1}{P_3\bar{\gamma}_3} \right) \right\} & \\
\quad + \frac{P_3\bar{\gamma}_3\log_2(e)}{P_3\bar{\gamma}_3 - P_1\bar{\gamma}_1}\left\{ 1 - \frac{1}{P_3\bar{\gamma}_3} e^{\frac{1}{P_3\bar{\gamma}_3}} E_1\left( \frac{1}{P_3\bar{\gamma}_3} \right) + e^{\frac{1}{P_3\bar{\gamma}_3}} E_1\left( \frac{1}{P_3\bar{\gamma}_3} \right) \right\}, & \text{if } P_1\bar{\gamma}_1 \neq P_2\bar{\gamma}_2 = P_3\bar{\gamma}_3, \\[3mm]
\frac{\log_2(e)}{2}\left\{ 3 - \frac{1}{P_1\bar{\gamma}_1} + \left( \frac{1}{P_1^2\bar{\gamma}_1^2} - \frac{2}{P_1\bar{\gamma}_1} + 2 \right) e^{\frac{1}{P_1\bar{\gamma}_1}} E_1\left( \frac{1}{P_1\bar{\gamma}_1} \right) \right\}, & \text{if } P_1\bar{\gamma}_1 = P_2\bar{\gamma}_2 = P_3\bar{\gamma}_3.
\end{cases}
\tag{21}
$$

We can rewrite the top of Equation (21) as

$$
\begin{aligned}
\bar{C}_{3,jdt} = \;& \frac{\log_2\left(e\right)P_1^2\bar{\gamma}_1^2}{\left(P_1\bar{\gamma}_1 - P_2\bar{\gamma}_2\right)\left(P_1\bar{\gamma}_1 - P_3\bar{\gamma}_3\right)} e^{\frac{1}{P_1\bar{\gamma}_1}} E_1\left( \frac{1}{P_1\bar{\gamma}_1} \right) \\
& + \frac{\log_2\left(e\right)P_2^2\bar{\gamma}_2^2}{\left(P_2\bar{\gamma}_2 - P_1\bar{\gamma}_1\right)\left(P_2\bar{\gamma}_2 - P_3\bar{\gamma}_3\right)} e^{\frac{1}{P_2\bar{\gamma}_2}} E_1\left( \frac{1}{P_2\bar{\gamma}_2} \right) \\
& + \frac{\log_2\left(e\right)P_3^2\bar{\gamma}_3^2}{\left(P_3\bar{\gamma}_3 - P_1\bar{\gamma}_1\right)\left(P_3\bar{\gamma}_3 - P_2\bar{\gamma}_2\right)} e^{\frac{1}{P_3\bar{\gamma}_3}} E_1\left( \frac{1}{P_3\bar{\gamma}_3} \right).
\end{aligned}
\tag{22}
$$

To calculate the optimum CTP powers that maximize the ECC, differentiating Equation (22) with respect to $P_3^*$ and setting the derivative to zero, we obtain Equation (23).

$$\left\{ P_1^{*2\prime}\bar{\gamma}_1 - \frac{P_1^{*2}\bar{\gamma}_1\left(P_1^{*\prime}\bar{\gamma}_1 - P_2^{*\prime}\bar{\gamma}_2\right)}{P_1^*\bar{\gamma}_1 - P_2^*\bar{\gamma}_2} - \frac{P_1^{*2}\bar{\gamma}_1\left(P_1^{*\prime}\bar{\gamma}_1 - \bar{\gamma}_3\right)}{P_1^*\bar{\gamma}_1 - P_3^*\bar{\gamma}_3} - P_1^{*\prime} \right\}$$

$$\cdot \frac{\log_2(e)\,\bar{\gamma}_1}{\left(P_1^*\bar{\gamma}_1 - P_2^*\bar{\gamma}_2\right)\left(P_1^*\bar{\gamma}_1 - P_3^*\bar{\gamma}_3\right)} e^{\frac{1}{P_1^*\bar{\gamma}_1}} E_1\left(\frac{1}{P_1^*\bar{\gamma}_1}\right)$$

$$+ \left\{ P_2^{*2\prime}\bar{\gamma}_2 - \frac{P_2^{*2}\bar{\gamma}_2\left(P_2^{*\prime}\bar{\gamma}_2 - P_1^{*\prime}\bar{\gamma}_1\right)}{P_2^*\bar{\gamma}_2 - \tilde{P}_1^*\bar{\gamma}_1} - \frac{P_2^{*2}\bar{\gamma}_2\left(P_2^{*\prime}\bar{\gamma}_2 - \bar{\gamma}_3\right)}{P_2^*\bar{\gamma}_2 - P_3^*\bar{\gamma}_3} - P_2^{\prime} \right\}$$

$$\cdot \frac{\log_2(e)\,\bar{\gamma}_2}{\left(P_2^*\bar{\gamma}_2 - P_1^*\bar{\gamma}_1\right)\left(P_2^*\bar{\gamma}_2 - P_3^*\bar{\gamma}_3\right)} e^{\frac{1}{P_2^*\bar{\gamma}_2}} E_1\left(\frac{1}{P_2^*\bar{\gamma}_2}\right)$$

$$+ \left\{ 2P_3^*\bar{\gamma}_3 - \frac{P_3^{*2}\bar{\gamma}_3\left(\bar{\gamma}_3 - \tilde{P}_1^{*\prime}\bar{\gamma}_1\right)}{P_3^*\bar{\gamma}_3 - \tilde{P}_1^*\bar{\gamma}_1} - \frac{P_3^2\bar{\gamma}_3\left(\bar{\gamma}_3 - P_2^{*\prime}\bar{\gamma}_2\right)}{P_3^*\bar{\gamma}_3 - P_2^*\bar{\gamma}_2} - 1 \right\} \quad (23)$$

$$\cdot \frac{\log_2(e)\,\bar{\gamma}_3}{\left(P_3^*\bar{\gamma}_3 - P_1^*\bar{\gamma}_1\right)\left(P_3^*\bar{\gamma}_3 - P_2^*\bar{\gamma}_2\right)} e^{\frac{1}{P_3^*\bar{\gamma}_3}} E_1\left(\frac{1}{P_3^*\bar{\gamma}_3}\right)$$

$$+ \frac{\log_2(e)\,P_1^*P_1^{*\prime}\bar{\gamma}_1^2}{\left(P_1^*\bar{\gamma}_1 - P_2^*\bar{\gamma}_2\right)\left(P_1^*\bar{\gamma}_1 - P_3^*\bar{\gamma}_3\right)} + \frac{\log_2(e)\,P_2^*P_2^{*\prime}\bar{\gamma}_2^2}{\left(P_2^*\bar{\gamma}_2 - P_1^*\bar{\gamma}_1\right)\left(P_2^*\bar{\gamma}_2 - P_3^*\bar{\gamma}_3\right)}$$

$$+ \frac{\log_2(e)\,P_3^*\bar{\gamma}_3^2}{\left(P_3^*\bar{\gamma}_3 - P_1^*\bar{\gamma}_1\right)\left(P_3^*\bar{\gamma}_3 - P_2^*\bar{\gamma}_2\right)} = 0.$$

In Equation (23), $P_1^*$, $P_2^*$, and $P_3^*$ denote the respective optimum CTP powers to be allocated to CTP1, CTP2, and CTP3 for Co3P JDT; and $(\cdot)'$ denotes $d(\cdot)/dP_3^*$.

At this point, we again introduce a simple CTP power expression for use in practice. The solid lines in Figure 3a–c show $P_3^*$ versus $\bar{\gamma}_3$[dB], with $\bar{\gamma}_1$[dB] and $\bar{\gamma}_2$[dB] as parameters, in the condition $\bar{\gamma}_1 \geq \bar{\gamma}_2 \geq \bar{\gamma}_3$. In this study, the solid lines were obtained using Equation (23) by exhaustive search. From these solid lines, we see that $P_3^*$ is almost linear with respect to $\bar{\gamma}_3$[dB] over the corresponding Co3P JDT region. Given $\bar{\gamma}_1$ and $\bar{\gamma}_2$, to obtain an approximate value of $P_3^*$, we use an LLA CTP power expression connecting the CTP3 powers corresponding to two different values of $\bar{\gamma}_3$: the boundary point dividing the Co3P JDT mode and Co2P JDT mode by CTP1 and CTP2, and the settled point of $P_3^* = P_2^*$ at $\bar{\gamma}_3 = \bar{\gamma}_2$, in the condition $\bar{\gamma}_1 \geq \bar{\gamma}_2 \geq \bar{\gamma}_3$.

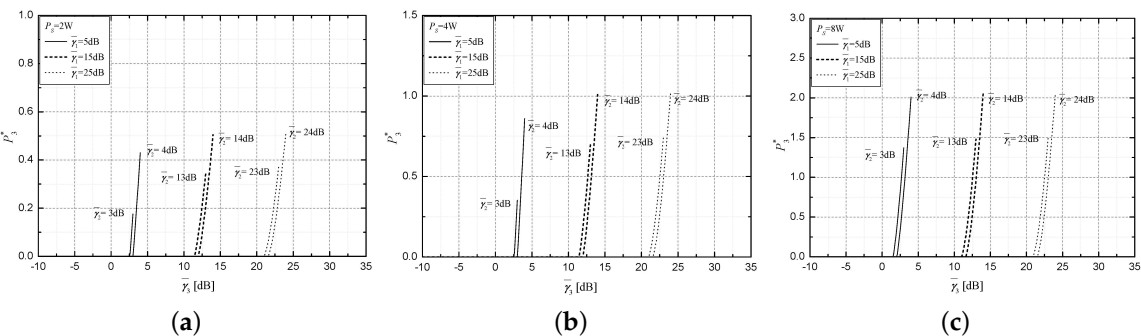

**Figure 3.** $P_3^*$ versus $\bar{\gamma}_3$[dB], with $\bar{\gamma}_1$[dB] and $\bar{\gamma}_2$[dB] as parameters: (**a**) $P_S = 2$ W, (**b**) $P_S = 4$ W, (**c**) $P_S = 8$ W.

If $P_3^*$ has been obtained, we have to calculate the corresponding optimum CTP powers for CTP1 and CTP2, (i.e., $P_1^*$ and $P_2^*$), which maximize the ECC. Now, we define two optimum CTP power functions with respect to $P_3^*$, with $\bar{\gamma}_1$, $\bar{\gamma}_2$, and $P_S$ as parameters, such as $\tilde{P}_1^*\left(\bar{\gamma}_1, \bar{\gamma}_2, P_S - P_3^*\right)$ and $\tilde{P}_2^*\left(\bar{\gamma}_1, \bar{\gamma}_2, P_S - P_3^*\right)$. In the following, for the notational convenience, we use simplified notations such as $\tilde{P}_1^*$ for $\tilde{P}_1^*\left(\bar{\gamma}_1, \bar{\gamma}_2, P_S - P_3^*\right)$ and $\tilde{P}_2^*$ for $\tilde{P}_2^*\left(\bar{\gamma}_1, \bar{\gamma}_2, P_S - P_3^*\right)$. Please note even though $P_3^*$ has been given, $\tilde{P}_1^*$ and $\tilde{P}_2^*$ could also be solved from Equation (23) only via exhaustive search. For the simplicity

of implementation, we can approximate $\tilde{P}_1^*$ and $\tilde{P}_2^*$ by invoking the LLA CTP power expression for $P_2^*$ versus $\bar{\gamma}_2[\text{dB}]$ in Equation (17), for Co2P JDT in the condition $\bar{\gamma}_1 \geq \bar{\gamma}_2$ as

$$\tilde{P}_1^* \approx \tilde{\beta}\left(\bar{\gamma}_1\right)\left(\ln \bar{\gamma}_1 - \ln \bar{\gamma}_2\right) + 0.5\left(P_S - P_3^*\right), \tag{24a}$$

$$\tilde{P}_2^* \approx \tilde{\beta}\left(\bar{\gamma}_1\right)\left(\ln \bar{\gamma}_2 - \ln \bar{\gamma}_1\right) + 0.5\left(P_S - P_3^*\right), \tag{24b}$$

where $\tilde{\beta}\left(\bar{\gamma}_1\right)$ at $\bar{\gamma}_2 = \bar{\gamma}_1$ is obtained by substituting the remaining power $P_S - P_3^*$ instead of $P_S$ into Equation (13) as Equation (25).

$$\tilde{\beta}(\bar{\gamma}_1) = \frac{-\frac{4}{\left(P_S-P_3^*\right)^2}e^{\frac{2}{\left(P_S-P_3^*\right)\bar{\gamma}_1}}E_1\left(\frac{2}{\left(P_S-P_3^*\right)\bar{\gamma}_1}\right) + \left(P_S-P_3^*\right)\bar{\gamma}_1^3 + \frac{2\bar{\gamma}_2}{P_S-P_1^*} - \bar{\gamma}_1^2}{\left\{\frac{24\bar{\gamma}_1}{\left(P_S-P_3^*\right)^2} + \frac{16}{\left(P_S-P_3^*\right)^3}\right\}e^{\frac{2}{\left(P_S-P_3^*\right)\bar{\gamma}_1}}E_1\left(\frac{2}{\left(P_S-P_3^*\right)\bar{\gamma}_1}\right) + 2\bar{\gamma}_1^3 - \frac{8\bar{\gamma}_1^2}{P_S-P_3^*} - \frac{8\bar{\gamma}_1}{\left(P_S-P_3^*\right)^2}}. \tag{25}$$

Now, we first obtain the boundary point dividing the Co3P JDT mode and Co2P JDT mode by CTP1 and CTP2. At the boundary condition of $P_3^*=0$, $\tilde{\beta}\left(\bar{\gamma}_1\right)$ can be rewritten as

$$\tilde{\beta}\left(\bar{\gamma}_1\right) = \frac{-\frac{4}{P_S^2}e^{\frac{2}{P_S\bar{\gamma}_1}}E_1\left(\frac{2}{P_S\bar{\gamma}_1}\right) + P_S\bar{\gamma}_1^3 + \frac{2\bar{\gamma}_1}{P_S} - \bar{\gamma}_1^2}{\left(\frac{24\bar{\gamma}_1}{P_S^2} + \frac{16}{P_S^3}\right)e^{\frac{2}{P_S\bar{\gamma}_1}}E_1\left(\frac{2}{P_S\bar{\gamma}_1}\right) + 2\bar{\gamma}_1^3 - \frac{8\bar{\gamma}_1^2}{P_S} - \frac{8\bar{\gamma}_1}{P_S^2}}. \tag{26}$$

Then, from (24), we can obtain

$$\tilde{P}_1^{*\prime} \approx -0.5, \tag{27a}$$

$$\tilde{P}_2^{*\prime} \approx -0.5, \tag{27b}$$

$$\begin{aligned}\tilde{P}_1^{*2} &\approx \tilde{\beta}(\bar{\gamma}_1)^2(\ln \bar{\gamma}_1 - \ln \bar{\gamma}_2)^2 + 0.25P_S^2 \\ &+ \tilde{\beta}\left(\bar{\gamma}_1\right)\left(\ln \bar{\gamma}_1 - \ln \bar{\gamma}_2\right)P_S,\end{aligned} \tag{27c}$$

$$\begin{aligned}\tilde{P}_2^{*2} &\approx \tilde{\beta}(\bar{\gamma}_1)^2(\ln \bar{\gamma}_2 - \ln \bar{\gamma}_1)^2 + 0.25P_S^2 \\ &+ \tilde{\beta}\left(\bar{\gamma}_1\right)\left(\ln \bar{\gamma}_2 - \ln \bar{\gamma}_1\right)P_S,\end{aligned} \tag{27d}$$

$$\tilde{P}_1^{*2\prime} \approx -0.5P_S - \tilde{\beta}\left(\bar{\gamma}_1\right)\left(\ln \bar{\gamma}_1 - \ln \bar{\gamma}_2\right), \tag{27e}$$

$$\tilde{P}_2^{*2\prime} \approx -0.5P_S - \tilde{\beta}\left(\bar{\gamma}_1\right)\left(\ln \bar{\gamma}_2 - \ln \bar{\gamma}_1\right). \tag{27f}$$

Substituting Equation (27) into Equation (23), after some manipulations, we obtain Equation (28) of the boundary point between the Co3P JDT mode and Co2P JDT mode by CTP1 and CTP2.

$$\bar{\gamma}_3^{bd,k} = \frac{N_{bd,k}}{D_{bd,k}}, \tag{28}$$

with

$$N_{bd,k} = \frac{\log_2(e)\,\bar{\gamma}_1}{\{\tilde{\beta}(\bar{\gamma}_1)(\ln\bar{\gamma}_1 - \ln\bar{\gamma}_2)(\bar{\gamma}_1 + \bar{\gamma}_2) + 0.5P_S(\bar{\gamma}_1 - \bar{\gamma}_2)\}\{\bar{\gamma}_1\tilde{\beta}(\bar{\gamma}_1)(\ln\bar{\gamma}_1 - \ln\bar{\gamma}_2) + 0.5\bar{\gamma}_1 P_S\}}$$

$$\cdot \left[ \bar{\gamma}_1\{-0.5P_S - \tilde{\beta}(\bar{\gamma}_1)(\ln\bar{\gamma}_1 - \ln\bar{\gamma}_2)\} - \frac{0.5\bar{\gamma}_1\{\tilde{\beta}(\bar{\gamma}_1)(\ln\bar{\gamma}_1 - \ln\bar{\gamma}_2) + 0.5P_S\}^2(\bar{\gamma}_2 - \bar{\gamma}_1)}{\tilde{\beta}(\bar{\gamma}_1)(\ln\bar{\gamma}_1 - \ln\bar{\gamma}_2)(\bar{\gamma}_1 + \bar{\gamma}_2) + 0.5P_S(\bar{\gamma}_1 - \bar{\gamma}_2)} \right.$$

$$\left. - \frac{\bar{\gamma}_1\{\tilde{\beta}(\bar{\gamma}_1)(\ln\bar{\gamma}_1 - \ln\bar{\gamma}_2) + 0.5P_S\}^2(-0.5\bar{\gamma}_1)}{\bar{\gamma}_1\tilde{\beta}(\bar{\gamma}_1)(\ln\bar{\gamma}_1 - \ln\bar{\gamma}_2) + 0.5\bar{\gamma}_1 P_S} + 0.5 \right]$$

$$\cdot e^{\frac{1}{\bar{\gamma}_1\tilde{\beta}(\bar{\gamma}_1)(\ln\bar{\gamma}_1 - \ln\bar{\gamma}_2) + 0.5\bar{\gamma}_1 P_S}} E_1\left( \frac{1}{\bar{\gamma}_1\tilde{\beta}(\bar{\gamma}_1)(\ln\bar{\gamma}_1 - \ln\bar{\gamma}_2) + 0.5\bar{\gamma}_1 P_S} \right)$$

$$+ \frac{\log_2(e)\,\bar{\gamma}_2}{\{\tilde{\beta}(\bar{\gamma}_1)(\ln\bar{\gamma}_2 - \ln\bar{\gamma}_1)(\bar{\gamma}_2 + \bar{\gamma}_1) + 0.5P_S(\bar{\gamma}_2 - \bar{\gamma}_1)\}\{\bar{\gamma}_2\tilde{\beta}(\bar{\gamma}_1)(\ln\bar{\gamma}_2 - \ln\bar{\gamma}_1) + 0.5\bar{\gamma}_2 P_S\}}$$

$$\cdot \left[ \bar{\gamma}_2\{-0.5P_S - \tilde{\beta}(\bar{\gamma}_1)(\ln\bar{\gamma}_2 - \ln\bar{\gamma}_1)\} - \frac{0.5\bar{\gamma}_2\{\tilde{\beta}(\bar{\gamma}_1)(\ln\bar{\gamma}_2 - \ln\bar{\gamma}_1) + 0.5P_S\}^2(\bar{\gamma}_1 - \bar{\gamma}_2)}{\tilde{\beta}(\bar{\gamma}_1)(\ln\bar{\gamma}_2 - \ln\bar{\gamma}_1)(\bar{\gamma}_2 + \bar{\gamma}_1) + 0.5P_S(\bar{\gamma}_2 - \bar{\gamma}_1)} \right.$$

$$\left. - \frac{\bar{\gamma}_2\{\tilde{\beta}(\bar{\gamma}_1)(\ln\bar{\gamma}_2 - \ln\bar{\gamma}_1) + 0.5P_S\}^2(-0.5\bar{\gamma}_2)}{\bar{\gamma}_2\tilde{\beta}(\bar{\gamma}_1)(\ln\bar{\gamma}_2 - \ln\bar{\gamma}_1) + 0.5\bar{\gamma}_2 P_S} + 0.5 \right]$$

$$\cdot e^{\frac{1}{\bar{\gamma}_2\tilde{\beta}(\bar{\gamma}_1)(\ln\bar{\gamma}_2 - \ln\bar{\gamma}_1) + 0.5\bar{\gamma}_2 P_S}} E_1\left( \frac{1}{\bar{\gamma}_2\tilde{\beta}(\bar{\gamma}_1)(\ln\bar{\gamma}_2 - \ln\bar{\gamma}_1) + 0.5\bar{\gamma}_2 P_S} \right)$$

$$+ \frac{-0.5\log_2(e)\,\bar{\gamma}_1^2\{\tilde{\beta}(\bar{\gamma}_1)(\ln\bar{\gamma}_1 - \ln\bar{\gamma}_2) + 0.5P_S\}}{\{\tilde{\beta}(\bar{\gamma}_1)(\ln\bar{\gamma}_1 - \ln\bar{\gamma}_2)(\bar{\gamma}_1 + \bar{\gamma}_2) + 0.5P_S(\bar{\gamma}_1 - \bar{\gamma}_2)\}\{\bar{\gamma}_1\tilde{\beta}(\bar{\gamma}_1)(\ln\bar{\gamma}_1 - \ln\bar{\gamma}_2) + 0.5\bar{\gamma}_1 P_S\}}$$

$$+ \frac{-0.5\log_2(e)\,\bar{\gamma}_2^2\{\tilde{\beta}(\bar{\gamma}_1)(\ln\bar{\gamma}_2 - \ln\bar{\gamma}_1) + 0.5P_S\}}{\{\tilde{\beta}(\bar{\gamma}_1)(\ln\bar{\gamma}_2 - \ln\bar{\gamma}_1)(\bar{\gamma}_2 + \bar{\gamma}_1) + 0.5P_S(\bar{\gamma}_2 - \bar{\gamma}_1)\}\{\bar{\gamma}_2\tilde{\beta}(\bar{\gamma}_1)(\ln\bar{\gamma}_2 - \ln\bar{\gamma}_1) + 0.5\bar{\gamma}_2 P_S\}},$$

$$D_{bd,k} = \frac{-\log_2(e)\,\bar{\gamma}_1^2\{\tilde{\beta}(\bar{\gamma}_1)(\ln\bar{\gamma}_1 - \ln\bar{\gamma}_2) + 0.5P_S\}^2}{\{\tilde{\beta}(\bar{\gamma}_1)(\ln\bar{\gamma}_1 - \ln\bar{\gamma}_2)(\bar{\gamma}_1 + \bar{\gamma}_2) + 0.5P_S(\bar{\gamma}_1 - \bar{\gamma}_2)\}\{\bar{\gamma}_1\tilde{\beta}(\bar{\gamma}_1)(\ln\bar{\gamma}_1 - \ln\bar{\gamma}_2) + 0.5\bar{\gamma}_1 P_S\}^2}$$

$$\cdot e^{\frac{1}{\bar{\gamma}_1\tilde{\beta}(\bar{\gamma}_1)(\ln\bar{\gamma}_1 - \ln\bar{\gamma}_2) + 0.5\bar{\gamma}_1 P_S}} E_1\left( \frac{1}{\bar{\gamma}_1\tilde{\beta}(\bar{\gamma}_1)(\ln\bar{\gamma}_1 - \ln\bar{\gamma}_2) + 0.5\bar{\gamma}_1 P_S} \right)$$

$$+ \frac{-\log_2(e)\,\bar{\gamma}_2^2\{\tilde{\beta}(\bar{\gamma}_1)(\ln\bar{\gamma}_2 - \ln\bar{\gamma}_1) + 0.5P_S\}^2}{\{\tilde{\beta}(\bar{\gamma}_1)(\ln\bar{\gamma}_2 - \ln\bar{\gamma}_1)(\bar{\gamma}_2 + \bar{\gamma}_1) + 0.5P_S(\bar{\gamma}_2 - \bar{\gamma}_1)\}\{\bar{\gamma}_2\tilde{\beta}(\bar{\gamma}_1)(\ln\bar{\gamma}_2 - \ln\bar{\gamma}_1) + 0.5\bar{\gamma}_2 P_S\}^2}$$

$$\cdot e^{\frac{1}{\bar{\gamma}_2\tilde{\beta}(\bar{\gamma}_1)(\ln\bar{\gamma}_2 - \ln\bar{\gamma}_1) + 0.5\bar{\gamma}_2 P_S}} E_1\left( \frac{1}{\bar{\gamma}_2\tilde{\beta}(\bar{\gamma}_1)(\ln\bar{\gamma}_2 - \ln\bar{\gamma}_1) + 0.5\bar{\gamma}_2 P_S} \right).$$

Next, we consider the settled point of $P_3^* = P_2^*$ at $\bar{\gamma}_3 = \bar{\gamma}_2$ in the condition $\bar{\gamma}_1 \geq \bar{\gamma}_2$ (or $\bar{\gamma}_1 \geq \bar{\gamma}_3$). To calculate $P_3^*$ that maximizes the ECC in the condition $P_3^* = P_2^*$ at $\bar{\gamma}_3 = \bar{\gamma}_2$, differentiating the middle of Equation (21) with respect to $P_3$ and setting the derivative to zero, we obtain Equation (29).

$$
\left\{ P_3^* \bar{\gamma}_3 - e^{\frac{1}{P_3^* \bar{\gamma}_3}} E_1\left(\frac{1}{P_3^* \bar{\gamma}_3}\right) + P_3^* \bar{\gamma}_3 e^{\frac{1}{P_3^* \bar{\gamma}_3}} E_1\left(\frac{1}{P_3^* \bar{\gamma}_3}\right) \right\}
$$

$$
\cdot \left( -P_3^* \bar{\gamma}_3 + P_1^* \bar{\gamma}_1 \right) \left( \bar{\gamma}_3 - \frac{dP_1^*}{dP_3^*} \bar{\gamma}_1 \right)
$$

$$
+ (P_3^* \bar{\gamma}_3 - P_1^* \bar{\gamma}_1)^2 \left\{ \frac{1}{P_3^{*2} \bar{\gamma}_3} e^{\frac{1}{P_3^* \bar{\gamma}_3}} E_1\left(\frac{1}{P_3^* \bar{\gamma}_3}\right) + \bar{\gamma}_3 e^{\frac{1}{P_3^* \bar{\gamma}_3}} E_1\left(\frac{1}{P_3^* \bar{\gamma}_3}\right) \right.
$$

$$
\left. - \frac{1}{P_3^*} e^{\frac{1}{P_3^* \bar{\gamma}_3}} E_1\left(\frac{1}{P_3^* \bar{\gamma}_3}\right) - \frac{1}{P_3^*} + 2\bar{\gamma}_3 \right\} - \frac{dP_1^*}{dP_3^*} \bar{\gamma}_1 \left( P_3^* \bar{\gamma}_3 - P_1^* \bar{\gamma}_1 \right)
$$

$$
\cdot \left\{ P_3^* \bar{\gamma}_3 e^{\frac{1}{P_3^* \bar{\gamma}_3}} E_1\left(\frac{1}{P_3^* \bar{\gamma}_3}\right) - P_1^* \bar{\gamma}_1 e^{\frac{1}{P_1^* \bar{\gamma}_1}} E_1\left(\frac{1}{P_1^* \bar{\gamma}_1}\right) \right\} \tag{29}
$$

$$
+ 2 P_1^* \bar{\gamma}_1 \left( \bar{\gamma}_3 - \frac{dP_1^*}{dP_3^*} \bar{\gamma}_1 \right) \left\{ P_3^* \bar{\gamma}_3 e^{\frac{1}{P_3^* \bar{\gamma}_3}} E_1\left(\frac{1}{P_3^* \bar{\gamma}_3}\right) - P_1^* \bar{\gamma}_1 e^{\frac{1}{P_1^* \bar{\gamma}_1}} E_1\left(\frac{1}{P_1^* \bar{\gamma}_1}\right) \right\}
$$

$$
- P_1^* \bar{\gamma}_1 (P_3^* \bar{\gamma}_3 - P_1^* \bar{\gamma}_1) \left\{ \bar{\gamma}_3 e^{\frac{1}{P_3^* \bar{\gamma}_3}} E_1\left(\frac{1}{P_3^* \bar{\gamma}_3}\right) - \frac{1}{P_3^*} e^{\frac{1}{P_3^* \bar{\gamma}_3}} E_1\left(\frac{1}{P_3^* \bar{\gamma}_3}\right) + \bar{\gamma}_3 \right.
$$

$$
\left. - \frac{dP_1^*}{dP_3^*} \bar{\gamma}_1 e^{\frac{1}{P_1^* \bar{\gamma}_1}} E_1\left(\frac{1}{P_1^* \bar{\gamma}_1}\right) + \frac{dP_1^*}{dP_3^*} \frac{1}{P_1^*} e^{\frac{1}{P_1^* \bar{\gamma}_1}} E_1\left(\frac{1}{P_1^* \bar{\gamma}_1}\right) - \frac{dP_1^*}{dP_3^*} \bar{\gamma}_1 \right\} = 0.
$$

At the settled point, we obtain $dP_1^*/dP_3^* = -2$, noting $P_1^* = P_S - 2P_3^*$. However, Equation (29) contains several complicated operations such as exponentials, exponential integrals, and their products. We can solve Equation (29) by only using numerical techniques such as the bisection method. The solid lines in Figure 4a–c show $P_3^*$ versus $\bar{\gamma}_3$[dB], with $\bar{\gamma}_1$[dB] as a parameter, when $P_S = 2, 4,$ and 8 W, respectively, in the condition $\bar{\gamma}_3 = \bar{\gamma}_2$. In this study, we obtained these solid lines from Equation (29) by using the bisection method. From these solid lines, we see that in the condition $\bar{\gamma}_1 \geq \bar{\gamma}_2 = \bar{\gamma}_3$, $P_3^*$ is almost quadratic with respect to $\bar{\gamma}_3$[dB] over the Co3P JDT region (i.e., the region from the non-JDT boundary point dividing the Co3P JDT mode and the non-JDT mode, to the balanced point of $P_1^* = P_2^* = P_3^*$ at $\bar{\gamma}_1 = \bar{\gamma}_2 = \bar{\gamma}_3$). Thus, we use the LLA CTP power expression for $P_3^*$ versus $\bar{\gamma}_3$[dB] in the condition $\bar{\gamma}_3 = \bar{\gamma}_2$, which is a linear line based on the linearization at the balanced point of $P_1^* = P_2^* = P_3^*$ at $\bar{\gamma}_1 = \bar{\gamma}_2 = \bar{\gamma}_3$. In addition, we can use an LQA CTP power expression for $P_3^*$ versus $\bar{\gamma}_3$[dB] in the condition $\bar{\gamma}_3 = \bar{\gamma}_2$, which is a quadratic line connecting the non-JDT boundary point of $P_2^* = P_3^* = 0$ at $\bar{\gamma}_3 = \bar{\gamma}_2$ and the balanced point of $P_1^* = P_2^* = P_3^*$ at $\bar{\gamma}_1 = \bar{\gamma}_2 = \bar{\gamma}_3$, while satisfying the slope at the balanced point.

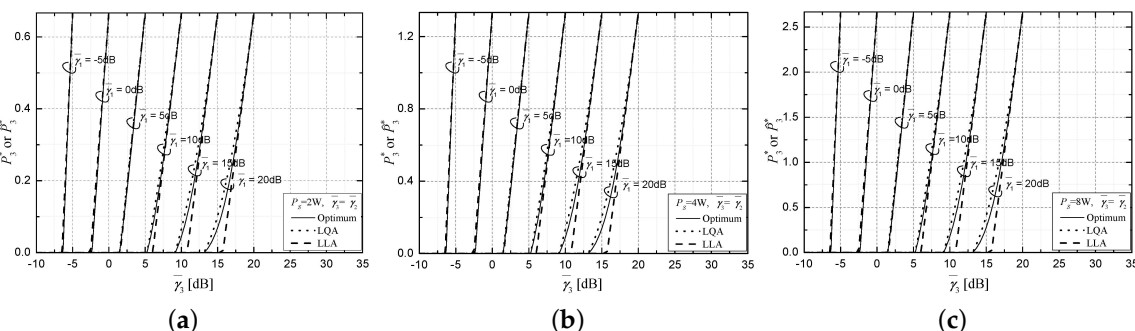

**Figure 4.** $P_3^*$ versus $\bar{\gamma}_3$ ($=\bar{\gamma}_2$)[dB], with $\bar{\gamma}_1$[dB] as a parameter: (**a**) $P_S = 2$ W, (**b**) $P_S = 4$ W, (**c**) $P_S = 8$ W.

We can approximate the optimum CTP3 power at the settled point of $P_3^* = P_2^*$ at $\bar{\gamma}_3 = \bar{\gamma}_2$, using the LLA or LQA CTP power expression. To derive the LQA expression in the condition $\bar{\gamma}_1 \geq \bar{\gamma}_2 = \bar{\gamma}_3$, we first derive the non-JDT boundary point. Given $\bar{\gamma}_1$, we obtain Equation (30) by substituting $P_3^* = 0$, $P_1^* = P_S - 2P_3^*$, and $dP_1^*/dP_3^* = -2$ into Equation (29).

$$
\begin{aligned}
&\lim_{P_3^* \to 0} \frac{1}{P_3^*} e^{\frac{1}{P_3^* \bar{\gamma}_3^{bd,e}}} E_1\left(\frac{1}{P_3^* \bar{\gamma}_3^{bd,e}}\right) - \lim_{P_3^* \to 0} \frac{P_S \bar{\gamma}_1}{P_3^{*2} \bar{\gamma}_3^{bd,e}} e^{\frac{1}{P_3^* \bar{\gamma}_3^{bd,e}}} E_1\left(\frac{1}{P_3^* \bar{\gamma}_3^{bd,e}}\right) \\
&+ \lim_{P_3^* \to 0} \frac{2P_S P_3^* \bar{\gamma}_1 \bar{\gamma}_3^{bd,e2}}{P_3^* \bar{\gamma}_3^{bd,e} - (P_S - 2P_3^*)\bar{\gamma}_1} e^{\frac{1}{P_3^* \bar{\gamma}_3^{bd,e}}} E_1\left(\frac{1}{P_3^* \bar{\gamma}_3^{bd,e}}\right) + \lim_{P_3^* \to 0} \frac{2\bar{\gamma}_1}{P_3^* \bar{\gamma}_3^{bd,e}} e^{\frac{1}{P_3^* \bar{\gamma}_3^{bd,e}}} E_1\left(\frac{1}{P_3^* \bar{\gamma}_3^{bd,e}}\right) \\
&+ \lim_{P_3^* \to 0} \frac{2P_S \bar{\gamma}_1}{P_3^*} e^{\frac{1}{P_3^* \bar{\gamma}_3^{bd,e}}} E_1\left(\frac{1}{P_3^* \bar{\gamma}_3^{bd,e}}\right) - \lim_{P_3^* \to 0} 2P_S \bar{\gamma}_1 \bar{\gamma}_3^{bd,e} e^{\frac{1}{P_3^* \bar{\gamma}_3^{bd,e}}} E_1\left(\frac{1}{P_3^* \bar{\gamma}_3^{bd,e}}\right) \\
&- \lim_{P_3^* \to 0} 2\bar{\gamma}_1 e^{\frac{1}{P_3^* \bar{\gamma}_3^{bd,e}}} E_1\left(\frac{1}{P_3^* \bar{\gamma}_3^{bd,e}}\right) + \lim_{P_3^* \to 0} \frac{P_S \bar{\gamma}_1}{P_3^*} + 2\bar{\gamma}_1 e^{\frac{1}{P_S \bar{\gamma}_1}} E_1\left(\frac{1}{P_S \bar{\gamma}_1}\right) \\
&+ 2P_S \bar{\gamma}_1 \bar{\gamma}_3^{bd,e} e^{\frac{1}{P_S \bar{\gamma}_1}} E_1\left(\frac{1}{P_S \bar{\gamma}_1}\right) - 2P_S \bar{\gamma}_1^2 - 3P_S \bar{\gamma}_1 \bar{\gamma}_3^{bd,e} - \bar{\gamma}_3^{bd,e} - 2\bar{\gamma}_1 = 0.
\end{aligned}
\tag{30}
$$

Invoking the Taylor series expansion at $P_3^* = 0$ for the third limiting term in Equation (30), we obtain Equation (31) [16].

$$
\begin{aligned}
&\lim_{P_3^* \to 0} \frac{P_S \bar{\gamma}_1}{P_3^{*2} \bar{\gamma}_3^{bd,e}} e^{\frac{1}{P_3^* \bar{\gamma}_3^{bd,e}}} E_1\left(\frac{1}{P_3^* \bar{\gamma}_3^{bd,e}}\right) = \\
&- \lim_{P_3^* \to 0} \frac{P_S \bar{\gamma}_1}{P_3^*} + P_S \bar{\gamma}_1 \bar{\gamma}_3^{bd,e} - \lim_{P_3^* \to 0} 2P_S P_3^* \bar{\gamma}_1 \bar{\gamma}_3^{bd,e2} + \lim_{P_3^* \to 0} O\left(P_3^{*2}\right).
\end{aligned}
\tag{31}
$$

Applying the limiting terms of Equation (8) and substituting Equation (31) into Equation (30), we obtain the non-JDT boundary point between the Co3P mode and the non-JDT mode along the settled points as follows:

$$
\bar{\gamma}_3^{bd,e} = \frac{P_S \bar{\gamma}_1 - e^{\frac{1}{P_S \bar{\gamma}_1}} E_1\left(\frac{1}{P_S \bar{\gamma}_1}\right)}{P_S e^{\frac{1}{P_S \bar{\gamma}_1}} E_1\left(\frac{1}{P_S \bar{\gamma}_1}\right)}.
\tag{32}
$$

Next, we derive the slope of $P_3^*$ with respect to $\bar{\gamma}_3[\mathrm{dB}]$ at the balanced point. For ease of derivation, we set $\alpha_3 = \ln(\bar{\gamma}_3)$. By differentiating Equation (29) with respect to $\alpha_3$, after some manipulations, we obtain Equation (33).

$$
\begin{aligned}
&\left(\frac{d}{d\alpha_3}\left(\frac{dP_1^*}{dP_3^*}\right)P_3^*\bar{\gamma}_1 e^{\alpha_3} + \frac{dP_1^*}{dP_3^*}\frac{dP_3^*}{d\alpha_3}\bar{\gamma}_1 e^{\alpha_3} + \frac{dP_1^*}{dP_3^*}P_3^*\bar{\gamma}_1 e^{\alpha_3}\right. \\
&-\frac{d}{d\alpha_3}\left(\frac{dP_1^*}{dP_3^*}\right)P_1^*\bar{\gamma}_1^2 - \frac{dP_1^*}{dP_3^*}\frac{dP_1^*}{d\alpha_3}\bar{\gamma}_1^2 - \frac{d}{d\alpha_3}\left(\frac{dP_1^*}{dP_3^*}\right)2P_3^{*2}\bar{\gamma}_1 e^{2\alpha_3} \\
&-\frac{dP_1^*}{dP_3^*}\frac{dP_3^*}{d\alpha_3}4P_3^*\bar{\gamma}_1 e^{2\alpha_3} - \frac{dP_1^*}{dP_3^*}4P_3^{*2}\bar{\gamma}_1 e^{2\alpha_3} - \frac{dP_3^*}{d\alpha_3}2P_3^* e^{2\alpha_3} \\
&-4P_3^* e^{2\alpha_3} + \frac{dP_3^*}{d\alpha_3}4P_3^* e^{3\alpha_3} + 6P_3^{*2}e^{3\alpha_3} + e^{\alpha_3} + \frac{dP_1^*}{d\alpha_3}4\bar{\gamma}_1 e^{\alpha_3} \\
&+4P_1^*\bar{\gamma}_1 e^{\alpha_3} - \frac{dP_1^*}{d\alpha_3}2P_3^*\bar{\gamma}_1 e^{2\alpha_3} - \frac{dP_3^*}{d\alpha_3}2P_1^*\bar{\gamma}_1 e^{2\alpha_3} - 4P_1^* P_3^*\bar{\gamma}_1 e^{2\alpha_3} \\
&-\frac{dP_1^*}{d\alpha_3}\frac{2\bar{\gamma}_1}{e^{\alpha_3}} + \frac{2P_1^*\bar{\gamma}_1}{e^{\alpha_3}} - \frac{dP_1^*}{d\alpha_3}\frac{2P_1^*\bar{\gamma}_1^2}{P_3^{*2}e^{\alpha_3}} \\
&\left.+\frac{dP_3^*}{d\alpha_3}\frac{2P_1^{*2}\bar{\gamma}_1^2}{P_3^{*3}e^{\alpha_3}} + \frac{P_1^{*2}\bar{\gamma}_1^2}{P_3^{*2}e^{\alpha_3}}\right) e^{\frac{1}{P_3^* e^{\alpha_3}}} E_1\left(\frac{1}{P_3^* e^{\alpha_3}}\right) \\
&+\left(\frac{dP_1^*}{dP_3^*}P_3^*\bar{\gamma}_1 e^{\alpha_3} - \frac{dP_1^*}{dP_3^*}P_1^*\bar{\gamma}_1^2 - \frac{dP_1^*}{dP_3^*}2P_3^{*2}\bar{\gamma}_1 e^{2\alpha_3}\right. \\
&-2P_3^* e^{2\alpha_3} + 2P_3^{*2}e^{3\alpha_3} + e^{\alpha_3} + 4P_1^*\bar{\gamma}_1 e^{\alpha_3} \\
&\left.-2P_1^* P_3^*\bar{\gamma}_1 e^{2\alpha_3} - \frac{2P_1^*\bar{\gamma}_1}{e^{\alpha_3}} - \frac{P_1^{*2}\bar{\gamma}_1^2}{P_3^{*2}e^{\alpha_3}}\right) \\
&\cdot\left\{-\left(\frac{dP_3^*}{d\alpha_3}\frac{1}{P_3^{*2}e^{\alpha_3}} + \frac{1}{P_3^* e^{\alpha_3}}\right)e^{\frac{1}{P_3^* e^{\alpha_3}}}E_1\left(\frac{1}{P_3^* e^{\alpha_3}}\right) + \left(\frac{dP_3^*}{d\alpha_3}\frac{1}{P_3^*}+1\right)\right\} \\
&+\left\{\frac{d}{d\alpha_3}\left(\frac{dP_1^*}{dP_3^*}\right)2P_1^* P_3^*\bar{\gamma}_1^2 e^{\alpha_3} + \frac{dP_1^*}{dP_3^*}\frac{dP_1^*}{d\alpha_3}2P_3^*\bar{\gamma}_1^2 e^{\alpha_3}\right. \\
&+\frac{dP_1^*}{dP_3^*}\frac{dP_3^*}{d\alpha_3}2P_1^*\bar{\gamma}_1^2 e^{\alpha_3} + \frac{dP_1^*}{dP_3^*}2P_1^* P_3^*\bar{\gamma}_1^2 e^{\alpha_3} + \frac{d}{d\alpha_3}\left(\frac{dP_1^*}{dP_3^*}\right)P_1^*\bar{\gamma}_1^2 \\
&+\frac{dP_1^*}{dP_3^*}\frac{dP_1^*}{d\alpha_3}\bar{\gamma}_1^2 - \frac{d}{d\alpha_3}\left(\frac{dP_1^*}{dP_3^*}\right)P_3^*\bar{\gamma}_1 e^{\alpha_3} - \frac{dP_1^*}{dP_3^*}\frac{dP_3^*}{d\alpha_3}\bar{\gamma}_1 e^{\alpha_3} \\
&\left.-\frac{dP_1^*}{dP_3^*}P_3^*\bar{\gamma}_1 e^{\alpha_3} - \frac{dP_1^*}{d\alpha_3}4P_1^*\bar{\gamma}_1^2 e^{\alpha_3} - 2P_1^{*2}\bar{\gamma}_1^2 e^{\alpha_3}\right\} e^{\frac{1}{P_1^*\bar{\gamma}_1}}E_1\left(\frac{1}{P_1^*\bar{\gamma}_1}\right) \\
&+\left(\frac{dP_1^*}{dP_3^*}2P_1^* P_3^*\bar{\gamma}_1^2 e^{\alpha_3} + \frac{dP_1^*}{dP_3^*}P_1^*\bar{\gamma}_1^2 - \frac{dP_1^*}{dP_3^*}P_3^*\bar{\gamma}_1 e^{\alpha_3} - 2P_1^{*2}\bar{\gamma}_1^2 e^{\alpha_3}\right) \\
&\cdot\left\{-\frac{dP_1^*}{d\alpha_3}\frac{1}{P_1^{*2}\bar{\gamma}_1}e^{\frac{1}{P_1^*\bar{\gamma}_1}}E_1\left(\frac{1}{P_1^*\bar{\gamma}_1}\right) + \frac{dP_1^*}{d\alpha_3}\frac{1}{P_1^*}\right\} \\
&+\frac{d}{d\alpha_3}\left(\frac{dP_1^*}{dP_3^*}\right)2P_1^* P_3^*\bar{\gamma}_1^2 e^{\alpha_3} + \frac{dP_1^*}{dP_3^*}\frac{dP_1^*}{d\alpha_3}2P_3^*\bar{\gamma}_1^2 e^{\alpha_3} + \frac{dP_1^*}{dP_3^*}\frac{dP_3^*}{d\alpha_3}2P_1^*\bar{\gamma}_1^2 e^{\alpha_3} \\
&+\frac{dP_1^*}{dP_3^*}2P_1^* P_3^*\bar{\gamma}_1^2 e^{\alpha_3} - \frac{d}{d\alpha_3}\left(\frac{dP_1^*}{dP_3^*}\right)P_1^{*2}e^{3\alpha_3} - \frac{dP_1^*}{dP_3^*}\frac{dP_1^*}{d\alpha_3}2P_1^* e^{3\alpha_3} \\
&-\frac{dP_1^*}{dP_3^*}3P_1^{*2}e^{3\alpha_3} - \frac{d}{d\alpha_3}\left(\frac{dP_1^*}{dP_3^*}\right)P_3^{*2}\bar{\gamma}_1 e^{2\alpha_3} - \frac{dP_1^*}{dP_3^*}\frac{dP_3^*}{d\alpha_3}2P_3^*\bar{\gamma}_1 e^{2\alpha_3} \\
&-\frac{dP_1^*}{dP_3^*}2P_3^{*2}\bar{\gamma}_1 e^{2\alpha_3} + \frac{dP_3^*}{d\alpha_3}6P_3^* e^{3\alpha_3} + 9P_3^{*2}e^{3\alpha_3} - \frac{dP_1^*}{d\alpha_3}6P_3^*\bar{\gamma}_1 e^{2\alpha_3} \\
&-\frac{dP_3^*}{d\alpha_3}6P_1^*\bar{\gamma}_1 e^{2\alpha_3} - 12P_1^* P_3^*\bar{\gamma}_1 e^{2\alpha_3} - \frac{dP_3^*}{d\alpha_3}e^{2\alpha_3} - 2P_3^* e^{2\alpha_3} + \frac{dP_1^*}{d\alpha_3}2\bar{\gamma}_1 e^{\alpha_3} \\
&+2P_1^*\bar{\gamma}_1 e^{\alpha_3} - \frac{dP_1^*}{d\alpha_3}2P_1^*\bar{\gamma}_1^2 e^{\alpha_3} - P_1^{*2}\bar{\gamma}_1^2 e^{\alpha_3} + \frac{dP_1^*}{d\alpha_3}\frac{2P_1^*\bar{\gamma}_1^2}{P_3^*} - \frac{dP_3^*}{d\alpha_3}\frac{P_1^{*2}\bar{\gamma}_1^2}{P_3^{*2}} = 0.
\end{aligned}
\tag{33}
$$

Please note $P_1^* = P_S - 2P_3^*$, $dP_1^*/dP_3^* = -2$, and $d\left(dP_1^*/dP_3^*\right)/d\alpha_3 = 0$ at the settled point of $P_3^* = P_2^*$ at $\bar{\gamma}_3 = \bar{\gamma}_2$. As mentioned above, assuming $P_3^*$, $P_1^*$ is a function with respect to $P_3^*$, with $\bar{\gamma}_1$, $\bar{\gamma}_2$, and $P_S$ as parameters, that is, $P_1^* = \tilde{P}_1^*\left(\bar{\gamma}_1, \bar{\gamma}_2, P_S - P_3^*\right)$. At the settled point of $\bar{\gamma}_3 = \bar{\gamma}_2$, $P_1^*$ can be also rewritten as $P_1^* = \tilde{P}_1^*\left(\bar{\gamma}_1, \bar{\gamma}_3, P_S - P_3^*\right)$ with respect to $\bar{\gamma}_3$ instead of $\bar{\gamma}_2$. For the simplicity of implementation, again invoking the LLA CTP power expression of Equation (17) and also noting $P_1^* = (P_S - P_3^*) - P_2^*$ in the condition $\bar{\gamma}_1 \geq \bar{\gamma}_2 = \bar{\gamma}_3$, we use the following approximate expression for $P_1^*$ as

$$\tilde{P}_1^* \approx \bar{\beta}\left(\bar{\gamma}_1\right)\left(\ln \bar{\gamma}_1 - \ln \bar{\gamma}_3\right) + 0.5\left(P_S - P_3^*\right), \tag{34}$$

with

$$\bar{\beta}\left(\bar{\gamma}_1\right) = \frac{-\frac{9}{P_S^2}e^{\frac{3}{P_S\bar{\gamma}_1}}E_1\left(\frac{3}{P_S\bar{\gamma}_1}\right) + \frac{2}{3}P_S\bar{\gamma}_1^3 + \frac{3\bar{\gamma}_1}{P_S} - \bar{\gamma}_1^2}{\left\{\frac{54\bar{\gamma}_1}{P_S^2} + \frac{54}{P_S^3}\right\}e^{\frac{3}{P_S\bar{\gamma}_1}}E_1\left(\frac{3}{P_S\bar{\gamma}_1}\right) + 2\bar{\gamma}_1^3 - \frac{12\bar{\gamma}_1^2}{P_S} - \frac{18\bar{\gamma}_1}{P_S^2}},$$

at the balanced point of $P_3^* = P_S/3$. Then, we can obtain

$$\frac{d\tilde{P}_1^*}{dP_3^*} \approx -0.5, \tag{35a}$$

$$\frac{d}{d\alpha_3}\left(\frac{d\tilde{P}_1^*}{dP_3^*}\right) \approx 0, \tag{35b}$$

$$\frac{d\tilde{P}_1^*}{d\alpha_3} \approx \bar{\beta}. \tag{35c}$$

Substituting Equation (35) into Equation (33), after some manipulations, we obtain Equation (36) of the approximate slope $\chi$ of $P_3^*$ with respect to $\alpha_3$ at the balanced point.

$$\chi = \lim_{\bar{\gamma}_3 \to \bar{\gamma}_1, P_3^* \to P_S/3} \frac{dP_3^*}{d\alpha_3} = \frac{N_\chi}{D_\chi}, \tag{36}$$

with

$$
\begin{aligned}
N_\chi = {} & \left\{\frac{64\bar{\beta}\left(\bar{\gamma}_1\right)}{P_S^3} + \frac{64}{P_S^4\bar{\gamma}_1} + \frac{16\bar{\beta}\left(\bar{\gamma}_1\right)}{P_S^2} + \frac{32}{P_S^3\bar{\gamma}_1} + \frac{4}{P_S^2\bar{\gamma}_1}\right\}e^{\frac{3}{P_S\bar{\gamma}_1}}E_1\left(\frac{3}{P_S\bar{\gamma}_1}\right) \\
& - \frac{16\bar{\gamma}_1}{P_S^2} - \frac{32\bar{\beta}\left(\bar{\gamma}_1\right)}{P_S^3} - \frac{16\bar{\beta}\left(\bar{\gamma}_1\right)}{P_S^2} - 4\bar{\gamma}_1^2 - P_S\bar{\gamma}_1^2 + \bar{\gamma}_1 - \frac{2\bar{\beta}\left(\bar{\gamma}_1\right)}{P_S},
\end{aligned}
$$

$$
\begin{aligned}
D_\chi = {} & -\left\{\frac{384\bar{\beta}\left(\bar{\gamma}_1\right)}{P_S^4} + \frac{256}{P_S^5\bar{\gamma}_1} + \frac{128\bar{\beta}\left(\bar{\gamma}_1\right)}{P_S^3} + \frac{128}{P_S^4\bar{\gamma}_1} + \frac{8\bar{\beta}\left(\bar{\gamma}_1\right)}{P_S^2} + \frac{16}{P_S^3\bar{\gamma}_1}\right\}e^{\frac{3}{P_S\bar{\gamma}_1}}E_1\left(\frac{3}{P_S\bar{\gamma}_1}\right) \\
& + \frac{128\bar{\beta}\left(\bar{\gamma}_1\right)}{P_S^4} + \frac{128\bar{\gamma}_1}{P_S^3} - \frac{32\bar{\gamma}_1^2}{P_S^2} + \frac{64\bar{\beta}\left(\bar{\gamma}_1\right)}{P_S^3} + \frac{32\bar{\gamma}_1}{P_S^2} + \frac{8\bar{\beta}\left(\bar{\gamma}_1\right)}{P_S^2} + 2\bar{\gamma}_1^2.
\end{aligned}
$$

To obtain the LQA CTP power expression for $P_3^*$ versus $\bar{\gamma}_3$[dB] along the settled points of $P_3^* = P_2^*$ at $\bar{\gamma}_3 = \bar{\gamma}_2$, we express a quadratic equation for $P_3^*$ versus $\bar{\gamma}_3$[dB] as $P_3^{q,sp*} = A_{sp}\bar{\gamma}_3$[dB]$^2 + B_{sp}\bar{\gamma}_3$[dB]$ + C_{sp}$, which must meet the three conditions of $P_3^{q,sp*} = 0$ at $\bar{\gamma}_3 = \bar{\gamma}_3^{bd,e}$ (i.e., the boundary point condition), $P_3^{q,sp*} = P_S/3$ at $\bar{\gamma}_1 = \bar{\gamma}_2 = \bar{\gamma}_3$ (i.e., the balanced point condition), and the slope at the balanced point $\chi$. From these three conditions, we obtain three simultaneous equations with respect to $A_{sp}$, $B_{sp}$, and $C_{sp}$, as

$$A_{sp}\bar{\gamma}_3^{bd,e}[\text{dB}]^2 + B_{sp}\bar{\gamma}_3^{bd,e}[\text{dB}] + C_{sp} = 0, \tag{37a}$$

$$A_{sp}\bar{\gamma}_1[\text{dB}]^2 + B_{sp}\bar{\gamma}_1[\text{dB}] + C_{sp} = P_S/3, \tag{37b}$$

$$2A_{sp}\bar{\gamma}_1[\text{dB}] + B_{sp} = \chi. \tag{37c}$$

Solving these three simultaneous equations, we obtain Equation (38) of the LQA CTP power expression for $P_3^*$ versus $\bar{\gamma}_3[\text{dB}]$ along the settled points.

$$
\begin{aligned}
P_3^{q,sp*} &= A_{sp}\bar{\gamma}_3[\text{dB}]^2 + B_{sp}\bar{\gamma}_3[\text{dB}] + C_{sp} \\
&= \left\{ \frac{\chi}{\left(\bar{\gamma}_1[\text{dB}] - \bar{\gamma}_3^{bd,e}[\text{dB}]\right)} - \frac{P_S}{3\left(\bar{\gamma}_1[\text{dB}] - \bar{\gamma}_3^{bd,e}[\text{dB}]\right)^2} \right\} \bar{\gamma}_3[\text{dB}]^2 \\
&\quad + \left\{ \chi - \frac{2\chi\bar{\gamma}_1[\text{dB}]}{\left(\bar{\gamma}_1[\text{dB}] - \bar{\gamma}_3^{bd,e}[\text{dB}]\right)} + \frac{2P_S\bar{\gamma}_1[\text{dB}]}{3\left(\bar{\gamma}_1[\text{dB}] - \bar{\gamma}_3^{bd,e}[\text{dB}]\right)^2} \right\} \bar{\gamma}_3[\text{dB}] \\
&\quad - \left\{ \frac{\chi}{\left(\bar{\gamma}_1[\text{dB}] - \bar{\gamma}_3^{bd,e}[\text{dB}]\right)} - \frac{P_S}{3\left(\bar{\gamma}_1[\text{dB}] - \bar{\gamma}_3^{bd,e}[\text{dB}]\right)^2} \right\} \bar{\gamma}_3^{bd,e}[\text{dB}]^2 \\
&\quad - \left\{ \chi - \frac{2\chi\bar{\gamma}_1[\text{dB}]}{\left(\bar{\gamma}_1[\text{dB}] - \bar{\gamma}_3^{bd,e}[\text{dB}]\right)} + \frac{2P_S\bar{\gamma}_1[\text{dB}]}{3\left(\bar{\gamma}_1[\text{dB}] - \bar{\gamma}_3^{bd,e}[\text{dB}]\right)^2} \right\} \bar{\gamma}_3^{bd,e}[\text{dB}].
\end{aligned}
\tag{38}
$$

Furthermore, using $\chi$, we obtain the LLA CTP power expression for $P_3^*$ versus $\bar{\gamma}_3[\text{dB}]$ along the settled points as

$$P_3^{l,sp*} = \chi\left(\ln\bar{\gamma}_3 - \ln\bar{\gamma}_1\right) + \frac{P_S}{3}. \tag{39}$$

In Figure 4a–c, the dotted and dashed lines, respectively, indicate the CTP3 powers when the LQA and LLA CTP power expressions are used, in the condition $\bar{\gamma}_3 = \bar{\gamma}_2$. From Figure 4a–c, we see that the CTP3 power using the LQA CTP power expression is closer to the corresponding optimum CTP3 power over the Co3P JDT region than that using the LLA CTP power expression.

Then, we can obtain the LLA CTP power expression for $P_3^*$ versus $\bar{\gamma}_3[\text{dB}]$, connecting the corresponding CTP3 powers at the boundary point between the Co3P JDT mode and Co2P JDT mode by CTP1 and CTP2, and at the settled point of $P_3^* = P_2^*$ at $\bar{\gamma}_3 = \bar{\gamma}_2$, as

$$P_3^{l,o*} = \frac{P_3^{q,sp*}}{\bar{\gamma}_1[\text{dB}] - \bar{\gamma}_3^{bd,k}[\text{dB}]}\left(\bar{\gamma}_3[\text{dB}] - \bar{\gamma}_3^{bd,k}[\text{dB}]\right), \tag{40a}$$

or

$$P_3^{l,o*} = \frac{P_3^{l,sp*}}{\bar{\gamma}_1[\text{dB}] - \bar{\gamma}_3^{bd,k}[\text{dB}]}\left(\bar{\gamma}_3[\text{dB}] - \bar{\gamma}_3^{bd,k}[\text{dB}]\right). \tag{40b}$$

Finally, in the condition $\bar{\gamma}_1 \geq \bar{\gamma}_2 \geq \bar{\gamma}_3$, we calculate the CTP3 power to be actually allocated to CTP3, $\hat{P}_3^*$ by using one of the LLA CTP3 power expressions of Equations (40a) and (40b) as

$$\hat{P}_3^* = \begin{cases} 0, & \text{if } P_3^{l,o*} < 0, \\ P_3^{l,o*}, & \text{if } P_3^{l,o*} \geq 0. \end{cases} \tag{41}$$

Substituting $\hat{P}_3^*$ obtained from Equation (41) instead of $P_3^*$ into Equation (24b), we can obtain the LLA CTP2 power expression for Co3P JDT as

$$P_2^{l,o*} = \tilde{\beta}\left(\bar{\gamma}_1\right)\left(\ln\bar{\gamma}_2 - \ln\bar{\gamma}_1\right) + 0.5\left(P_S - \hat{P}_3^*\right). \tag{42}$$

We calculate the CTP2 power to be actually allocated to CTP2, $\hat{P}_2^*$ by using the LLA CTP2 power expression of Equation (42) instead of Equation (24b) as

$$\hat{P}_2^* = \begin{cases} 0, & \text{if } P_2^{l,o*} < 0, \\ P_2^{l,o*}, & \text{if } P_2^{l,o*} \geq 0. \end{cases} \tag{43}$$

Subsequently, we can obtain the CTP1 power to be actually allocated to CTP1, $\hat{P}_1^*$ as $\hat{P}_1^* = P_S - \hat{P}_2^* - \hat{P}_3^*$.

In conclusion, we can decide whether Co3P JDT or Co2P JDT is performed or not by using Equation (41) for Co3P JDT or Equations (41) and (43) for Co2P JDT. That is, in the condition $\bar{\gamma}_1 \geq \bar{\gamma}_2 \geq \bar{\gamma}_3$, Co3P JDT is performed if $\hat{P}_3^* > 0$; Co2P JDT by CTP1 and CTP2 is performed if $\hat{P}_3^* = 0$ and $\hat{P}_2^* > 0$; and single-point transmission by CTP1 is performed if $\hat{P}_3^* = 0$ and $\hat{P}_2^* = 0$.

## 5. Numerical Results

We evaluated the ECCs of the Co2P JDT network using the LQA and LLA CTP power expressions and the ECCs of the Co3P JDT network using the LLA CTP power expressions. For Co3P JDT, we use the LQA CTP power expression for $P_3^*$ versus $\bar{\gamma}_3$[dB] along the settled points, i.e., Equation (38), to obtain the approximate CTP3 power along the settled points.

Figure 5a–c show the ECCs of the Co2P JDT network as a function of $\bar{\gamma}_2$[dB], with $\bar{\gamma}_1$[dB] as a parameter, when $P_S = 2$, 4, and 8 W, respectively. From Figure 5a–c, we see that the ECCs are constant when $\bar{\gamma}_2$ is much smaller than $\bar{\gamma}_1$ because the ECCs are mainly determined by $\bar{\gamma}_1$. On the contrary, we see that the ECCs are mainly determined by $\bar{\gamma}_2$ when $\bar{\gamma}_2$ is much greater than $\bar{\gamma}_1$. When $\bar{\gamma}_2$[dB] is at and around the balanced point, we can use Co2P JDT. For $P_S = 2$, 4, and 8 W and $\bar{\gamma}_1 = -5$, 0, 5, 10, 15, and 20 dB, the approximate ECC curves using the LQA CTP power expression in Equation (16) have maximum and average percent errors of 0.0196% and 0.0017%, respectively, while the approximate ECC curves using the LLA CTP power expression in Equation (18) have maximum and average percent errors of 0.3818% and 0.0114%, respectively. From Figure 5a–c, we see that the Co2P JDT region becomes wider and the average percent error increases, as $\bar{\gamma}_1$ increases. We also see that the Co2P JDT region becomes wider while the average percent error is almost steady, as $P_S$ increases.

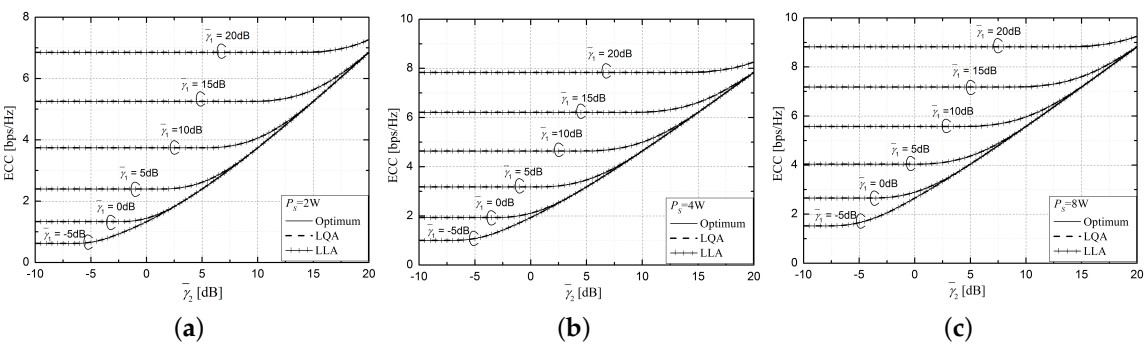

**Figure 5.** ECCs of the Co2P JDT network as a function of $\bar{\gamma}_2$[dB], with $\bar{\gamma}_1$[dB] as a parameter: (**a**) $P_S = 2$ W, (**b**) $P_S = 4$ W, (**c**) $P_S = 8$ W.

Figure 6a–c show the ECCs of the Co3P JDT network as a function of $\bar{\gamma}_3$[dB], with $\bar{\gamma}_1$[dB] and $\bar{\gamma}_2$[dB] as parameters, over the regime of $\bar{\gamma}_3 \leq \bar{\gamma}_2$, when $P_S = 2$, 4, and 8 W, respectively. From Figure 6a, the approximate ECC curves using the LLA CTP power expression in Equation (40a) have maximum and average percent errors of 0.0194% and 0.0079%, respectively. From Figure 6a–c, we see that the Co3P JDT region becomes wider, although the average percent error is almost steady around 0.01%, as $\bar{\gamma}_1$ and/or $P_S$ increase.

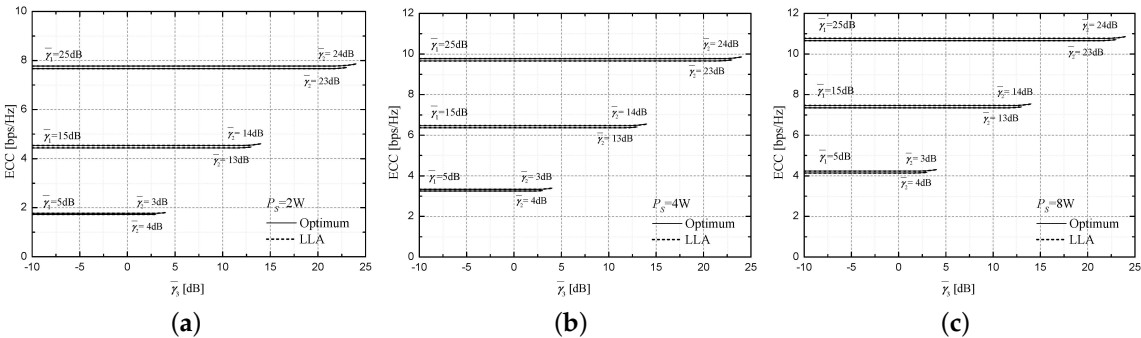

**Figure 6.** ECCs of the Co3P JDT network as a function of $\bar{\gamma}_3$[dB], with $\bar{\gamma}_1$[dB] and $\bar{\gamma}_2$[dB] as parameters: (**a**) $P_S$ = 2 W, (**b**) $P_S$ = 4 W, (**c**) $P_S$ = 8 W.

In conclusion, the approximated ECCs using the LLA and LQA CTP power expressions for Co2P JDT and the LLA CTP power expressions for Co3P JDT trace the optimum ECCs well with no noticeable difference.

Figure 7a–c show the ECCs achieved by the Co3P JDT network, the Co2P JDT network by CTP1 and CTP2, and the non-JDT network by CTP1 as a function of the mean branch GNR difference in decibels, when $\bar{\gamma}_1$ = 5 dB and $P_S$ = 2, 4, and 8 W, respectively. Denoting the mean branch GNR difference in dB as $\Delta\gamma$[dB], $\bar{\gamma}_2$[dB] = $\bar{\gamma}_1$[dB]$-\Delta\gamma$[dB] and $\bar{\gamma}_3$[dB] = $\bar{\gamma}_2$[dB]$-\Delta\gamma$[dB] = $\bar{\gamma}_1$[dB]$-2\Delta\gamma$[dB]. From Figure 7a, for $P_S$ = 2 W, the ECC of the Co3P JDT network is higher than the ECC of the Co2P JDT network over the Co3P JDT region, and the ECC of the Co2P JDT network is higher than the ECC of the non-JDT network over the Co2P JDT region. As expected, the difference between the ECC of the Co3P JDT network and the ECC of the Co2P JDT network is the largest at the balanced point of $\bar{\gamma}_1 = \bar{\gamma}_2 = \bar{\gamma}_3$, that is, when the mean branch GNR difference in dB is zero. The difference in the ECC becomes smaller as $\Delta\gamma$[dB] increases. When $P_S$ = 2 W and $\bar{\gamma}_1$ = 5 dB at the balanced point of $\bar{\gamma}_1 = \bar{\gamma}_2 = \bar{\gamma}_3$, the ECC of the Co3P JDT network is 3.1% higher than that of the Co2P JDT network and 12.4% higher than that of the non-JDT single-branch network. From the above results, we see that the CoMP JDT network can obtain the maximum diversity gain at the balanced point. From Figure 7a–c, we see that the ECC of the CoMP JDT increases, and the Co3P and Co2P JDT regions become wider, as $P_S$ increases. We also confirm that the ECC of CoMP JDT increases and the CoMP JDT region becomes wider, as $\bar{\gamma}_1$ increases, even though we have not demonstrated these results here. When $P_S$ increases from 2 W to 4 W, and $P_S$ increases from 4 W to 8 W, the corresponding Co3P and Co2P JDT regions become wider by 0.2 dB and 0.5 dB, respectively.

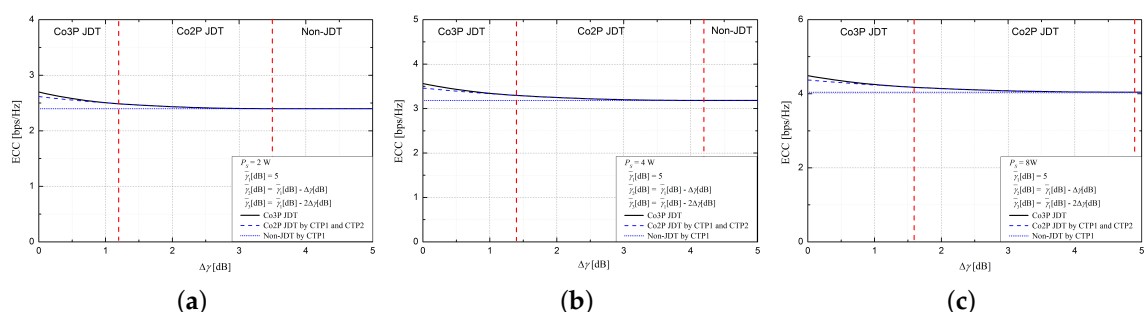

**Figure 7.** ECCs versus $\Delta\gamma$[dB] ($\bar{\gamma}_1$ = 5 dB): (**a**) $P_S$ = 2 W, (**b**) $P_S$ = 4 W, (**c**) $P_S$ = 8 W.

## 6. Conclusions

In this paper, we presented an efficient JPA method for the CoMP JDT network, aimed at maximizing the ECC in Rayleigh fading channels. In the Co2P JDT network, we introduced the LQA and LLA CTP power expressions for obtaining the optimum CTP powers. In the Co3P JDT network,

we introduced a simple LLA CTP power expression for obtaining the optimum power of the worst quality-providing CTP. After obtaining the approximate CTP power for the worst quality-providing CTP, we obtained the respective LLA CTP power expressions for the two better quality-providing CTPs by invoking the LLA CTP power expressions for Co2P JDT, under the remaining power given by the TCPP minus the CTP power for the worst quality-providing CTP. From the numerical results, we verified that the LQA and LLA CTP power expressions for Co2P JDT and the LLA CTP power expressions for Co3P JDT are very efficient in terms of the simplicity for JPA and CoMP set selection, as well as the resulting ECC. When $P_S$ = 2, 4, and 8 W and $\bar{\gamma}_2$ = −5, 0, 5, 10, 15 and 20 dB, the approximate ECC curves of the Co2P JDT network, using the LQA CTP power expression have maximum and average percent errors of 0.0196% and 0.0017%, respectively, whereas the approximate ECC curves using the LLA CTP power expression have maximum and average percent errors of 0.3818% and 0.0114%, respectively. When $P_S$ = 2, 4, and 8 W, the approximate ECC curves of the Co3P JDT network using the LLA CTP power expression have maximum and average percent errors of 0.0194% and 0.0079%, respectively. In conclusion, the proposed JDT power allocation method can efficiently determine how many CTPs participate in JDT and simply and efficiently determine the CTP power levels to be allocated over the CTPs in the resulting JDT cooperating set.

**Author Contributions:** Conceptualization, M.L. and S.-K.O.; methodology, M.L. and S.-K.O.; software, M.L.; validation, M.L. and S.-K.O.; formal analysis, M.L. and S.-K.O.; investigation, M.L. and S.-K.O.; resources, M.L. and S.-K.O.; data curation, M.L.; writing-original draft preparation, M.L.; writing-review and editing, S.-K.O.; visualization, M.L.; supervision, S.-K.O.

**Funding:** This research received no external funding.

**Conflicts of Interest:** The authors declare no conflict of interest.

## Abbreviations

The following abbreviations are used in this manuscript:

| | |
|---|---|
| 4G | fourth-generation |
| Co2P | coordinated two-point |
| Co3P | coordinated three-point |
| CoMP | coordinated multi-point |
| CTP | coordinated transmission point |
| ECC | ergodic cooperative capacity |
| GNR | gain-to-noise ratio |
| JDT | joint diversity transmission |
| JPA | joint power allocation |
| LLA | log-linear approximated |
| LQA | log-quadratic approximated |
| TCPP | total coordination point power |
| UE | user equipment |

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
