# Peer review of "Joint Power Allocation for Coordinated Multi-Point Diversity Transmission in Rayleigh Fading Channels"

_electronics, doi:10.3390/electronics8010101_

Round 1

Reviewer 1 Report

This paper proposes a joint power allocation method for CoMP JDT network to maximize the ergodic cooperative capacity in Rayleigh fading channels. The CTP power expressions are given. The topic considered in this paper, joint power allocation, is very interesting. However, there are some problems need to be addressed before being considered for publication.

1.     If the number of CTP is larger than 3, how to joint allocate the power efficiently? The LQA and LLA CTP power expressions for Co2P JDT power allocation and the LQA CTP power expression for Co3P JDT power allocation are derived and given in this paper. Does the method given in this paper for power allocation still valid for a more general situation (for example, the number of CTP is M>3)?

2.     If the channel parameter changes fast, for example, fast fading channels, does the method still valid? Will calculation of CTP power expressions (time and accuracy) affect the performance?  How does it affect the performance?

Author Response

Response to Reviewer 1 Comments

This paper proposes a joint power allocation method for CoMP JDT network to maximize the ergodic cooperative capacity in Rayleigh fading channels. The CTP power expressions are given. The topic considered in this paper, joint power allocation, is very interesting. However, there are some problems need to be addressed before being considered for publication.

Point 1:  If the number of CTP is larger than 3, how to joint allocate the power efficiently? The LQA and LLA CTP power expressions for Co2P JDT power allocation and the LQA CTP power expression for Co3P JDT power allocation are derived and given in this paper. Does the method given in this paper for power allocation still valid for a more general situation (for example, the number of CTP is M>3)?

Response 1: For the network including larger than 3 CTPs, optimum joint power allocation is very complicated, but the improvement on ergodic cooperative capacity is imperceptible. In a practical network including larger than 3, it can be extremely efficient to select 2 or 3 CTPs with the best quality channel gain, and then to allocate joint powers for the selected 2 or 3 CTPs using the results of this work.

Point 2:If the channel parameter changes fast, for example, fast fading channels, does the method still valid? Will calculation of CTP power expressions (time and accuracy) affect the performance?  How does it affect the performance?

Response 2: The results of this work use means of channel gains, not instantaneous channel gains. Hence, it is more effective for fast fading channels. If we get accurate statistics of fast fading channel gains, we can allocate optimum joint powers without performance degradation.

Reviewer 2 Report

The authors consider an important problem in cellular networks - power control in cooperative networks. This work contains a generalization of the author's previous work on 2 point network to 3 point network.

The contribution will be more clear if the authors explain more clearly the significance and non-triviality of the 2 to 3 generalization. In particular, can the results be further generalized to more points? 

The technical results rely a lot on the previous work and could be explained in more detail to make this work easier to understand, e.g., the capacity formulas in (4).

For the references, please list all authors (avoiding using et al, e.g., see [2]).

Author Response

Response to Reviewer 2 Comments

The authors consider an important problem in cellular networks - power control in cooperative networks. This work contains a generalization of the author's previous work on 2 point network to 3 point network.

Point 1: The contribution will be more clear if the authors explain more clearly the significance and non-triviality of the 2 to 3 generalization. In particular, can the results be further generalized to more points? 

Response 1: We appreciate your good comment. We have added some significance and non-triviality of the 2 to 3 generalization in light of your comment.

Point 2: The technical results rely a lot on the previous work and could be explained in more detail to make this work easier to understand, e.g., the capacity formulas in (4).

Response 2: We have added explanations of the formulas in (4).

Point 3: For the references, please list all authors (avoiding using et al, e.g., see [2]).

Response 3: We have listed all authors for the references.

Round 2

Reviewer 1 Report

There are still two problems need to be addressed before being considered for publication.

1.     Does the method proposed in this paper for power allocation valid for the general situation, for example, the number of CTP is M>3?

2.     If the channel parameter changes fast, for example, fast fading channels, does the method still valid?

Author Response

Response to Reviewer 1 Comments (Round 2)

There are still two problems need to be addressed before being considered for publication.

Point 1: Does the method proposed in this paper for power allocation valid for the general situation, for example, the number of CTP is M>3?

Response 1: The proposed method is not valid for power allocation for the general number of CTPs. It is only valid for the situation of 2 or 3 CTPs. In a practical system including larger than 3 CTPs, there is a high probability that the system selects 2 or 3 CTPs with better quality of average channel gains, and then allocates joint powers over the selected 2 or 3 CTPs using the proposed method.

Point 2: If the channel parameter changes fast, for example, fast fading channels, does the method still valid?

Response 2:

In the proposed method, joint power levels over CTPs are calculated using the average channel gains obtained from statistics for varying fading channels. Hence, the proposed method is valid for fast fading channels. We are not allocate power levels over CTPs using instantaneous gains of varying fading channels, but allocate joint power levels over CTPs using average gains of varying fading channels in the proposed method. If the system produce accurate statistics for fast fading channel gains, we can allocate optimum joint powers over CTPs by using the proposed method without performance degradation.

Round 3

Reviewer 1 Report

The authors addressed all my questions in their response. I have no further comments.